# Doubly Mild Generalization for Offline Reinforcement Learning

**Yixiu Mao**[1], **Qi Wang**[1], **Yun Qu**[1], **Yuhang Jiang**[1], **Xiangyang Ji**[1]
[1]Department of Automation, Tsinghua University
myx21@mails.tsinghua.edu.cn, xyji@tsinghua.edu.cn

## Abstract

Offline Reinforcement Learning (RL) suffers from the extrapolation error and value overestimation. From a generalization perspective, this issue can be attributed to the over-generalization of value functions or policies towards out-of-distribution (OOD) actions. Significant efforts have been devoted to mitigating such generalization, and recent in-sample learning approaches have further succeeded in entirely eschewing it. Nevertheless, we show that mild generalization beyond the dataset can be trusted and leveraged to improve performance under certain conditions. To appropriately exploit generalization in offline RL, we propose Doubly Mild Generalization (DMG), comprising (i) mild action generalization and (ii) mild generalization propagation. The former refers to selecting actions in a close neighborhood of the dataset to maximize the Q values. Even so, the potential erroneous generalization can still be propagated, accumulated, and exacerbated by bootstrapping. In light of this, the latter concept is introduced to mitigate the generalization propagation without impeding the propagation of RL learning signals. Theoretically, DMG guarantees better performance than the in-sample optimal policy in the oracle generalization scenario. Even under worst-case generalization, DMG can still control value overestimation at a certain level and lower bound the performance. Empirically, DMG achieves state-of-the-art performance across Gym-MuJoCo locomotion tasks and challenging AntMaze tasks. Moreover, benefiting from its flexibility in both generalization aspects, DMG enjoys a seamless transition from offline to online learning and attains strong online fine-tuning performance.

## 1   Introduction

Reinforcement learning (RL) aims to solve sequential decision-making problems and has garnered significant attention in recent years [53, 67, 74, 63, 12]. However, its practical applications encounter several challenges, such as risky exploration attempts [20] and time-consuming data collection phases [35]. Offline RL emerges as a promising paradigm to alleviate these challenges by learning without interaction with the environment [40, 42]. It eliminates the need for unsafe exploration and facilitates the utilization of pre-existing large-scale datasets [31, 48, 59].

However, offline RL suffers from the out-of-distribution (OOD) issue and extrapolation error [19]. From a generalization perspective, this well-known challenge can be regarded as a consequence of the over-generalization of value functions or policies towards OOD actions [47]. Specifically, the potential value over-estimation at OOD actions caused by intricate generalization is often improperly captured by the max operation [73]. This over-estimation will propagate to values of in-distribution samples through Bellman backups and further spread to values of OOD ones via generalization. In mitigating value overestimation caused by OOD actions, substantial efforts have been dedicated [19, 39, 38, 17] and recent advancements in in-sample learning have successfully formulated the Bellman target solely with the actions present in the dataset [37, 85, 92, 88, 21] and extracted policies by weighted

behavior cloning [57, 80]. As a result, these algorithms completely eschew generalization and avoid the extrapolation error. Despite simplicity, this way can not take advantage of the generalization ability of neural networks, which could be beneficial for performance improvement. Until now, how to appropriately exploit generalization in offline RL remains a lasting issue.

This work demonstrates that mild generalization beyond the dataset can be trusted and leveraged to improve performance under certain conditions. For appropriate exploitation of mild generalization, we propose Doubly Mild Generalization (DMG) for offline RL, comprising (i) mild action generalization and (ii) mild generalization propagation. The former concept refers to choosing actions in the vicinity of the dataset to maximize the Q values. However, the mere utilization of mild action generalization still falls short in adequately circumventing potential erroneous generalization, which can be propagated, accumulated, and exacerbated through the process of bootstrapping. To address this, we propose a novel concept, mild generalization propagation, which involves reducing the generalization propagation while preserving the propagation of RL learning signals. Regarding DMG's implementation, this work presents a simple yet effective scheme. Specifically, we blend the mildly generalized max with the in-sample max in the Bellman target, where the former is achieved by actor-critic learning with regularization towards high-value in-sample actions, and the latter is accomplished using in-sample learning techniques such as expectile regression [37].

We conduct a thorough theoretical analysis of our approach DMG in both oracle and worst-case generalization scenarios. Under oracle generalization, DMG guarantees better performance than the in-sample optimal policy in the dataset [38, 37]. Even under worst-case generalization, DMG can still upper bound the overestimation of value functions and guarantee to output a safe policy with a performance lower bound. Empirically[1], DMG achieves state-of-the-art performance on standard offline RL benchmarks [16], including Gym-MuJoCo locomotion tasks and challenging AntMaze tasks. Moreover, benefiting from its flexibility in both generalization aspects, DMG can seamlessly transition from offline to online learning and attain superior online fine-tuning performance.

## 2 Preliminaries

**RL.**  The environment in RL is mostly characterized as a Markov decision process (MDP), which can be represented as a tuple $\mathcal{M} = (\mathcal{S}, \mathcal{A}, P, R, \gamma, d_0)$, comprising the state space $\mathcal{S}$, action space $\mathcal{A}$, transition dynamics $P : \mathcal{S} \times \mathcal{A} \to \Delta(\mathcal{S})$, reward function $R : \mathcal{S} \times \mathcal{A} \to [0, R_{\max}]$, discount factor $\gamma \in [0, 1)$, and initial state distribution $d_0$ [70]. The goal of RL is to find a policy $\pi : \mathcal{S} \to \Delta(\mathcal{A})$ that can maximize the expected discounted return, denoted as $J(\pi)$:

$$J(\pi) = \mathbb{E}_{s_0 \sim d_0, a_t \sim \pi(\cdot|s_t), s_{t+1} \sim P(\cdot|s_t, a_t)} \left[ \sum_{t=0}^{\infty} \gamma^t R(s_t, a_t) \right]. \tag{1}$$

For any policy $\pi$, we define the value function as $V^\pi(s) = \mathbb{E}_\pi \left[ \sum_{t=0}^{\infty} \gamma^t R(s_t, a_t) | s_0 = s \right]$ and the state-action value function (Q-value function) as $Q^\pi(s, a) = \mathbb{E}_\pi \left[ \sum_{t=0}^{\infty} \gamma^t R(s_t, a_t) | s_0 = s, a_0 = a \right]$.

**Offline RL.**  Distinguished from traditional online RL training, offline RL handles a static dataset of transitions $\mathcal{D} = \{(s_i, a_i, r_i, s_i')\}_{i=0}^{n-1}$ and seeks an optimal policy without any additional data collection [40, 42]. We use $\hat{\beta}(a|s)$ to denote the empirical behavior policy observed in $\mathcal{D}$, which depicts the conditional distributions in the dataset [19]. Ordinary approximate dynamic programming methods minimize temporal difference error, according to the following loss [70]:

$$L_{TD}(\theta) = \mathbb{E}_{(s,a,s') \sim \mathcal{D}} \left[ (Q_\theta(s, a) - R(s, a) - \gamma \max_{a'} Q_{\theta'}(s', a'))^2 \right], \tag{2}$$

where $\pi_\phi$ is a parameterized policy, $Q_\theta(s, a)$ is a parameterized $Q$ function, and $Q_{\theta'}(s, a)$ is a target $Q$ function whose parameters are updated via Polyak averaging [53].

## 3 Doubly Mild Generalization for Offline RL

This section discusses the strategy to appropriately exploit generalization in offline RL. In Section 3.1, we introduce a formal perspective on how generalization impacts offline RL and discuss the issues of

---

[1]Our code is available at https://github.com/maoyixiu/DMG.

over-generalization and non-generalization. Subsequently, we propose the DMG concept, comprising mild action generalization and mild generalization propagation in Section 3.2. Following this, we conduct a comprehensive analysis of DMG in both oracle generalization (Section 3.3) and worst-case generalization scenarios (Section 3.4). Finally, we present the practical algorithm in Section 3.5.

## 3.1 Generalization Issues in Offline RL

Offline RL training typically involves a complex interaction between Bellman backup and generalization [47]. Offline RL algorithms vary in backup mechanisms to train the Q function. Here we denote a generic form of Bellman backup as $\mathcal{T}_u$, where $u$ is a distribution in the action space.

$$\mathcal{T}_u Q(s, a) := R(s, a) + \gamma \mathbb{E}_{s' \sim P(\cdot|s,a)} \left[ \max_{a' \sim u(\cdot|s')} Q(s', a') \right] \tag{3}$$

During offline training, this backup is exclusively executed on $(s, a) \in \mathcal{D}$, and the values of $(s, a) \notin \mathcal{D}$ are influenced solely via generalization. A crucial aspect is that $(s', a')$ in the Bellman target can be absent from the dataset $\mathcal{D}$, depending on the choice of $u$. As a result, Bellman backup and generalization exhibit an intricate interaction: the backups on $(s, a) \in \mathcal{D}$ impact the values of $(s, a) \notin \mathcal{D}$ via generalization; the values of $(s, a) \notin \mathcal{D}$ participates in the computation of Bellman target, thereby affecting the values of $(s, a) \in \mathcal{D}$.

This interaction poses a key challenge in offline RL, value overestimation. The potential overestimation of values of $(s, a) \notin \mathcal{D}$, induced by intricate generalization, tends to be improperly captured by the max operation, a phenomenon known as maximization bias [73]. This overestimation propagates to values of $(s, a) \in \mathcal{D}$ through backups and further extends to values of $(s, a) \notin \mathcal{D}$ via generalization. This cyclic process consistently amplifies value overestimation, potentially resulting in value divergence. The crux of this detrimental process can be summarized as **over-generalization**.

To address value overestimation, recent advancements in the field have introduced a paradigm known as in-sample learning, which formulates the Bellman target solely with the actions present in the dataset [37, 85, 92, 88, 21]. Its effect is equivalent to choosing $u$ in $\mathcal{T}_u$ to be exactly $\hat{\beta}$, i.e., the empirical behavior policy observed in the dataset. Following in-sample value learning, policies are extracted from the learned Q functions using weighted behavior cloning [57, 9, 55]. By entirely eschewing generalization in offline RL training, they effectively avoid the extrapolation error [19], a strategy we term **non-generalization**. However, the ability to generalize is a critical factor contributing to the extensive utilization of neural networks [41]. In this sense, in-sample learning methods seem too conservative without utilizing generalization, particularly when the offline datasets do not cover the optimal actions in large or continuous spaces.

## 3.2 Doubly Mild Generalization

The following focuses on the appropriate exploitation of generalization in offline RL.

We start by analyzing the generalization effect under the generic backup operator $\mathcal{T}_u$. We consider a straightforward scenario, where $Q_\theta$ is updated to $Q_{\theta'}$ by one gradient step on a single $(s, a) \in \mathcal{D}$ with learning rate $\alpha$. We characterize the resulting generalization effect on any $(s, \tilde{a}) \notin \mathcal{D}$[2] as follows.

**Theorem 1** (Informal). *Under certain continuity conditions, the following equation holds when the learning rate $\alpha$ is sufficiently small and $\tilde{a}$ is sufficiently close to $a$:*

$$Q_{\theta'}(s, \tilde{a}) = Q_\theta(s, \tilde{a}) + C_1 \left( \mathcal{T}_u Q_\theta(s, \tilde{a}) - Q_\theta(s, \tilde{a}) + C_2 \|\tilde{a} - a\| \right) + \mathcal{O}\left( \|\theta' - \theta\|^2 \right) \tag{4}$$

*where $C_1 \in [0, 1]$ and $C_2$ is a bounded constant.*

The formal theorem and all proofs are deferred to Appendix B.

Note that Eq. (4) is the update of the parametric Q function ($Q_\theta \rightarrow Q_{\theta'}$) at state-action pairs $(s, \tilde{a}) \notin \mathcal{D}$, which is exclusively caused by generalization. If $\tilde{a}$ is within a close neighborhood of $a$, then $C_2 \|\tilde{a} - a\|$ is small. Moreover, as $C_1 \in [0, 1]$, Eq. (4) approximates an update towards the true objective $\mathcal{T}_u Q_\theta(s, \tilde{a})$, as if $Q_\theta(s, \tilde{a})$ is updated by a true gradient step at $(s, \tilde{a}) \notin \mathcal{D}$. Therefore,

---

[2]The interplay between backup and generalization does not involve states out of the dataset (Bellman target does not contain OOD states), hence we do not consider $(\tilde{s}, \tilde{a}) \notin \mathcal{D}$, though the analysis of $Q(\tilde{s}, \tilde{a})$ is similar.

Theorem 1 shows that, under certain continuity conditions, Q functions can generalize well and approximate true updates in a close neighborhood of samples in the dataset. This implies that mild generalizations beyond the dataset can be leveraged to potentially pursue better performance. Inspired by Theorem 1, we define a mildly generalized policy $\tilde{\beta}$ as follows.

**Definition 1** (Mildly generalized policy). *Policy $\tilde{\beta}$ is termed a mildly generalized policy if it satisfies*

$$\mathrm{supp}(\hat{\beta}(\cdot|s)) \subseteq \mathrm{supp}(\tilde{\beta}(\cdot|s)), \text{ and } \max_{a_1 \sim \tilde{\beta}(\cdot|s)} \min_{a_2 \sim \hat{\beta}(\cdot|s)} \|a_1 - a_2\| \leq \epsilon_a, \tag{5}$$

*where $\hat{\beta}$ is the empirical behavior policy observed in the offline dataset.*

It means that $\tilde{\beta}$ has a wider support than $\hat{\beta}$ (the dataset), and for any $a_1 \sim \tilde{\beta}(\cdot|s)$, we can find $a_2 \sim \hat{\beta}(\cdot|s)$ (in dataset) such that $\|a_1 - a_2\| \leq \epsilon_a$. In other words, the generalization of $\tilde{\beta}$ beyond the dataset is bounded by $\epsilon_a$ when measured in the action space distance. According to Theorem 1, there is a high chance that $Q_\theta$ can generalize well in this mild generalization area $\tilde{\beta}(a|s) > 0$.

However, even in this mild generalization area, it is inevitable that the learned value function will incur some degree of generalization error. The possible erroneous generalization can still be propagated and exacerbated by value bootstrapping as discussed in Section 3.1. To this end, we introduce an additional level of mild generalization, termed mild generalization propagation, and propose a novel Doubly Mildly Generalization (DMG) operator as follows.

**Definition 2.** *The Doubly Mild Generalization (DMG) operator is defined as*

$$\mathcal{T}_{\mathrm{DMG}} Q(s,a) := R(s,a) + \gamma \mathbb{E}_{s' \sim P(\cdot|s,a)} \left[ \lambda \max_{a' \sim \tilde{\beta}(\cdot|s')} Q(s',a') + (1-\lambda) \max_{a' \sim \hat{\beta}(\cdot|s')} Q(s',a') \right] \tag{6}$$

*where $\hat{\beta}$ is the empirical behavior policy in the dataset and $\tilde{\beta}$ is a mildly generalized policy.*

Note that in typical offline RL algorithms, extrapolation error and value overestimation caused by erroneous generalization are propagated through bootstrapping, and the discount factor of this process is $\gamma$. DMG reduces this discount factor to $\lambda\gamma$, mitigating the amplification of value overestimation. On the other hand, in contrast to in-sample methods, DMG allows mild generalization, utilizing the generalization ability of neural networks to seek better performance, as Theorem 1 suggests that value functions are highly likely to generalize well in the mild generalization area.

To summarize, the generalization of DMG is mild in two aspects: (i) **mild action generalization**: based on the mildly generalized policy $\tilde{\beta}$, which generalizes beyond $\hat{\beta}$, DMG selects actions in a close neighborhood of the dataset to maximize the Q values in the first part of the Bellman target; and (ii) **mild generalization propagation**: DMG mitigates the generalization propagation without hindering the propagation of RL learning signals by blending the mildly generalized max with the in-sample max in the Bellman target. This reduces the discount factor through which generalization propagates, mitigating the amplification of value overestimation caused by bootstrapping.

To support the above claims, we provide a comprehensive analysis of DMG in both oracle and worst-case generalization scenarios, with particular emphasis on value estimation and performance.

### 3.3 Oracle Generalization

This section conducts analyses under the assumption that the learned value functions can achieve oracle generalization in the mild generalization area $\tilde{\beta}(a|s) > 0$, formally defined as follows.

**Assumption 1** (Oracle generalization). *The generalization of learned Q functions in the mild generalization area $\tilde{\beta}(a|s) > 0$ reflects the true value updates according to $\mathcal{T}_{\mathrm{DMG}}$.*

The mild generalization area $\tilde{\beta}(a|s) > 0$ may contain some points outside the offline dataset, and $\mathcal{T}_{\mathrm{DMG}}$ might query Q values of such points. This assumption assumes that the generalization at such points reflects the true value updates according to $\mathcal{T}_{\mathrm{DMG}}$. The rationale for such an assumption comes from Theorem 1, which characterizes the generalization effect of value functions in the mild generalization area. Now we analyze the dynamic programming properties of the operators $\mathcal{T}_{\mathrm{DMG}}$ and $\mathcal{T}_{\mathrm{In}}$, where $\mathcal{T}_{\mathrm{In}}$ is the in-sample Q learning operator [37, 88, 21] defined as follows.

**Definition 3.** *The In-sample Q Learning operator [37] is defined as*

$$\mathcal{T}_{\text{In}}Q(s,a) := R(s,a) + \gamma \mathbb{E}_{s'\sim P(\cdot|s,a)} \left[ \max_{a'\sim\hat{\beta}(\cdot|s')} Q(s',a') \right] \tag{7}$$

*where $\hat{\beta}$ is the empirical behavior policy in the dataset.*

**Lemma 1.** *$\mathcal{T}_{\text{In}}$ is a $\gamma$-contraction operator in the in-sample area $\hat{\beta}(a|s) > 0$ under the $\mathcal{L}_\infty$ norm.*

Following Lemma 1, we denote the fixed point of $\mathcal{T}_{\text{In}}$ as $Q^*_{\text{In}}$, and its induced policy as $\pi^*_{\text{In}}$. Here $Q^*_{\text{In}}$ is known as the in-sample optimal value function [37], which is the value function of the in-sample optimal policy $\pi^*_{\text{In}}$. We refer readers to [37, 38, 88] for more discussions on the in-sample optimality.

Now we present the theoretical properties of DMG for comparison.

**Theorem 2** (Contraction). *Under Assumption 1, $\mathcal{T}_{\text{DMG}}$ is a $\gamma$-contraction operator in the mild generalization area $\tilde{\beta}(a|s) > 0$ under the $\mathcal{L}_\infty$ norm. Therefore, by repeatedly applying $\mathcal{T}_{\text{DMG}}$, any initial Q function can converge to the unique fixed point $Q^*_{\text{DMG}}$.*

We denote the induced policy of $Q^*_{\text{DMG}}$ as $\pi^*_{\text{DMG}}$, whose performance is guaranteed as follows.

**Theorem 3** (Performance). *Under Assumption 1, the value functions of $\pi^*_{\text{DMG}}$ and $\pi^*_{\text{In}}$ satisfy:*

$$V^{\pi^*_{\text{DMG}}}(s) \geq V^{\pi^*_{\text{In}}}(s), \quad \forall s \in \mathcal{D}. \tag{8}$$

Theorem 3 indicates that the policy learned by DMG can achieve better performance than the in-sample optimal policy under the oracle generalization condition.

### 3.4 Worst-case Generalization

This section turns to the analyses in the worst-case generalization scenario, where the learned value functions may exhibit poor generalization in the mild generalization area $\tilde{\beta}(a|s) > 0$. In other words, this section considers that $\mathcal{T}_{\text{DMG}}$ is only defined in the in-sample area $\hat{\beta}(a|s) > 0$ and the learned value functions may have any generalization error at other state-action pairs. In this case, we use the notation $\hat{\mathcal{T}}_{\text{DMG}}$ to tell the difference.

We make continuity assumptions about the learned Q function and the transition dynamics.

**Assumption 2** (Lipschitz Q). *The learned Q function is $K_Q$-Lipschitz. $\forall s \sim \mathcal{D}, \forall a_1, a_2 \sim \mathcal{A}$, $|Q(s,a_1) - Q(s,a_2)| \leq K_Q\|a_1 - a_2\|$*

**Assumption 3** (Lipschitz P). *The transition dynamics P is $K_P$-Lipschitz. $\forall s, s' \sim \mathcal{S}, \forall a_1, a_2 \sim \mathcal{A}$, $|P(s'|s,a_1) - P(s'|s,a_2)| \leq K_P\|a_1 - a_2\|$*

For Assumption 2, a continuous learned Q function is particularly necessary for analyzing value function generalization and can be relatively easily satisfied using neural networks or linear models [24]. Assumption 3 is also a common assumption in theoretical studies of RL [13, 87, 61].

Now we consider the iteration starting from arbitrary function $Q^0$: $\hat{Q}^k_{\text{DMG}} = \hat{\mathcal{T}}_{\text{DMG}}\hat{Q}^{k-1}_{\text{DMG}}$ and $Q^k_{\text{In}} = \mathcal{T}_{\text{In}}Q^{k-1}_{\text{In}}, \forall k \in \mathbb{Z}^+$. The possible value of $\hat{Q}^k_{\text{DMG}}$ is bounded by the following results.

**Theorem 4** (Limited overestimation). *Under Assumption 2, the learned Q function of DMG by iterating $\hat{\mathcal{T}}_{\text{DMG}}$ satisfies the following inequality*

$$Q^k_{\text{In}}(s,a) \leq \hat{Q}^k_{\text{DMG}}(s,a) \leq Q^k_{\text{In}}(s,a) + \frac{\lambda\epsilon_a K_Q\gamma}{1-\gamma}(1-\gamma^k), \ \forall s,a \sim \mathcal{D}, \ \forall k \in \mathbb{Z}^+. \tag{9}$$

Since in-sample training eliminates the extrapolation error [37, 92], $Q^k_{\text{In}}$ can be considered a relatively accurate estimate [37]. Therefore, Theorem 4 suggests that DMG exhibits limited value overestimation under the worst-case generalization scenario. Moreover, the bound becomes tighter as $\epsilon_a$ decreases (milder action generalization) and $\lambda$ decreases (milder generalization propagation). This is consistent with our intuitions in Section 3.2.

Finally, we show in Theorem 5 that even under worst-case generalization, DMG guarantees to output a safe policy with a performance lower bound.

**Theorem 5** (Performance lower bound). *Let $\hat{\pi}_{\mathrm{DMG}}$ be the learned policy of DMG by iterating $\hat{\mathcal{T}}_{\mathrm{DMG}}$, $\pi^*$ be the optimal policy, and $\epsilon_{\mathcal{D}}$ be the inherent performance gap of the in-sample optimal policy $\epsilon_{\mathcal{D}} := J(\pi^*) - J(\pi^*_{\mathrm{In}})$. Under Assumptions 2 and 3, for sufficiently small $\epsilon_a$, we have*

$$J(\hat{\pi}_{\mathrm{DMG}}) \geq J(\pi^*) - \frac{CK_P R_{\max}}{1 - \gamma}\epsilon_a - \epsilon_{\mathcal{D}}. \qquad (10)$$

*where $C$ is a positive constant.*

### 3.5 Practical Algorithm

This section puts DMG into implementation and presents a simple yet effective practical algorithm. The algorithm comprises the following networks: policy $\pi_\phi$, target policy $\pi_{\phi'}$, $Q$ network $Q_\theta$, target $Q$ network $Q_{\theta'}$, and $V$ network $V_\psi$.

**Policy learning.** Practically, we expect DMG to exhibit a tendency towards mild generalization around good actions in the dataset. To this end, we first consider reshaping the empirical behavior policy $\hat{\beta}$ to be skewed towards actions with high advantage values $\hat{\beta}^*(a|s) \propto \hat{\beta}(a|s)\exp(A(s,a))$. Then we enforce the proximity between the trained policy and the reshaped behavior policy to constrain the generalization area. We define the generalization set $\Pi_G$ as follows.

$$\Pi_G = \{\pi \mid \mathrm{KL}(\hat{\beta}^*(\cdot|s)\|\pi(\cdot|s)) \leq \epsilon\} \qquad (11)$$

Note that forward KL allows $\pi$ to select actions outside the support of $\hat{\beta}^*$, enabling $\Pi_G$ to generalize beyond the actions in the dataset. With $\Pi_G$ defined, the next step is to compute the maximal $Q$ within $\Pi_G$. To accomplish this, we adopt Actor-Critic style training [70] for this part.

$$\max_\phi \mathbb{E}_{s\sim\mathcal{D}, a\sim\pi_\phi(\cdot|s)}Q_\theta(s,a), \quad s.t. \ \pi_\phi \in \Pi_G \qquad (12)$$

By treating the constraint term as a penalty, we maximize the following objective.

$$\max_\phi \mathbb{E}_{s\sim\mathcal{D}, a\sim\pi_\phi(\cdot|s)}Q_\theta(s,a) - \nu\mathbb{E}_{s\sim\mathcal{D}}\mathrm{KL}(\hat{\beta}^*(\cdot|s)\|\pi_\phi(\cdot|s)) \qquad (13)$$

Through straightforward derivations, Eq. (13) is equivalent to the following policy training objective.

$$J_\pi(\phi) = \mathbb{E}_{s\sim\mathcal{D}, a\sim\pi_\phi(\cdot|s)}Q_\theta(s,a) - \nu\mathbb{E}_{(s,a)\sim\mathcal{D}}\left[\exp(\alpha(Q_{\theta'}(s,a) - V_\psi(s)))\log\pi_\phi(a|s)\right] \qquad (14)$$

where $\alpha$ is an inverse temperature and $Q_{\theta'}(s,a) - V_\psi(s)$ computes the advantage function $A(s,a)$.

**Value learning.** Now we turn to the implementation of the $\mathcal{T}_{\mathrm{DMG}}$ operator for training value functions. By introducing the aforementioned policy, we can substitute $\max_{a\sim\tilde{\beta}}$ in $\mathcal{T}_{\mathrm{DMG}}$ with $\mathbb{E}_{a\sim\pi}$. Regarding $\max_{a\sim\hat{\beta}}$ in $\mathcal{T}_{\mathrm{DMG}}$, any in-sample learning techniques can be employed to compute the in-sample maximum [37, 88, 85, 21]. In particular, based on IQL [37], we perform expectile regression.

$$L_V(\psi) = \mathbb{E}_{(s,a)\sim\mathcal{D}}\left[L_2^\tau\left(Q_{\theta'}(s,a) - V_\psi(s)\right)\right] \quad (15)$$

---
**Algorithm 1** DMG
---
1: Initialize $\pi_\phi$, $\pi_{\phi'}$, $Q_\theta$, $Q_{\theta'}$, and $V_\psi$.
2: **for** each gradient step **do**
3:     Update $\psi$ by minimizing Eq. (15)
4:     Update $\theta$ by minimizing Eq. (16)
5:     Update $\phi$ by maximizing Eq. (14)
6:     Update target networks: $\theta' \leftarrow (1-\xi)\theta' + \xi\theta$, $\phi' \leftarrow (1-\xi)\phi' + \xi\phi$
7: **end for**
---

where $L_2^\tau(u) = |\tau - \mathbb{1}(u < 0)|u^2$ and $\tau \in (0, 1)$. For $\tau \approx 1$, $V_\psi$ can capture the in-sample maximal $Q$ [37]. Finally, we have the following value training loss.

$$L_Q(\theta) = \mathbb{E}_{(s,a,s')\sim\mathcal{D}}\left[\left(Q_\theta(s,a) - R(s,a) - \gamma\lambda\mathbb{E}_{a'\sim\pi_{\phi'}}Q_{\theta'}(s',a') - \gamma(1-\lambda)V_\psi(s')\right)^2\right] \qquad (16)$$

**Overall algorithm.** Integrating all components, we present our practical algorithm in Algorithm 1.

# 4 Discussions and Related Work

**Summary of offline RL work from a generalization perspective.** As analyzed above, DMG is featured in both mild action generalization and mild generalization propagation. Within the actor-critic framework upon which most offline RL algorithms are built, these two aspects correspond to the policy and value training phases, respectively. Action generalization concerns whether the policy training intentionally selects actions beyond the dataset to maximize Q values, while generalization propagation involves whether value training propagates generalization through bootstrapping. Table 1 presents a clear comparison of offline RL works in this generalization view. The table shows one representative method of each category and we elaborate on others as follows.

Table 1: Comparison of offline RL work from the generalization perspective.

|  | IQL | AWAC | TD3BC | TD3 | DMG (Ours) |
|---|---|---|---|---|---|
| Action generalization | *none* | *none* | *mild* | *full* | *mild* |
| Generalization propagation | *none* | *full* | *full* | *full* | *mild* |

Concerning policy learning, AWR [57], AWAC [55], CRR [80], 10% BC [8], IQL [37], and other works such as [78, 9, 66, 21, 88] extract policies through weighted or filtered behavior cloning, thereby lacking intentional action generalization to maximize Q values beyond the dataset. Typical policy-regularized offline RL methods like TD3BC [17], BRAC [84], BEAR [38], SPOT [83], and others such as [79, 61, 72] introduce regularization terms to Q maximization objectives to regularize the trained policy towards the behavior policy and allows mild action generalization. Online RL algorithms like TD3 [18] and SAC [27] have no constraints and maximize Q values in the entire action space, corresponding to full action generalization. Regarding value training, in-sample learning methods including OneStep RL [7], IQL [37], InAC [85], IAC [92], $\mathcal{X}$QL [21], and SQL [88] completely avoid generalization propagation and accumulation via bootstrapping, whereas typical offline and online RL approaches allow full generalization propagation through bootstrapping. In the proposed approach DMG, generalization is mild in both aspects.

Recently, Ma et al. [47] have also drawn attention to generalization in offline RL and the issue of over-generalization. They mitigate over-generalization from a representation perspective, differentiating between the representations of in-sample and OOD state-action pairs. Lyu et al. [44] argue that conventional value penalization like CQL [39] tends to harm the generalization of value functions and hinder performance improvement. They propose mild value penalization to mitigate the detrimental effects of value penalization on generalization.

**Connection to heuristic blending approaches.** Our approach also relates to the framework of blending heuristics into bootstrapping [10, 81, 71, 28, 82, 22]. In offline RL, HUBL [22] blends Monte-Carlo returns into bootstrapping and acts as a data relabeling step, which reduces the degree of bootstrapping and thereby increases its performance. In contrast, DMG blends the in-sample maximal values into the bootstrapping operator. DMG does not reduce the discount for RL learning but reduces the discount for generalization propagation.

For extended discussions on related work, please refer to Appendix A.

# 5 Experiments

In this section, we conduct several experiments to justify the validity of the proposed method DMG. Experimental details and extended results are provided in Appendices C and D, respectively.

## 5.1 Main Results on Offline RL Benchmarks

**Tasks.** We evaluate the proposed approach on Gym-MuJoCo locomotion domains and challenging AntMaze domains in D4RL [16]. The latter involves sparse-reward tasks and necessitates "stitching" fragments of suboptimal trajectories traveling undirectedly to find a path to the goal of the maze.

**Baselines.** Our offline RL baselines include both typical bootstrapping methods and in-sample learning approaches. For the former, we compare to BCQ [19], BEAR [38], AWAC [55], TD3BC [17],

Table 2: Averaged normalized scores on Gym locomotion and Antmaze tasks over five random seeds. m = medium, m-r = medium-replay, m-e = medium-expert, e = expert, r = random; u = umaze, u-d = umaze-diverse, m-p = medium-play, m-d = medium-diverse, l-p= large-play, l-d = large-diverse.

| Dataset-v2 | BC | BCQ | BEAR | DT | AWAC | OneStep | TD3BC | CQL | IQL | DMG (Ours) |
|---|---|---|---|---|---|---|---|---|---|---|
| halfcheetah-m | 42.0 | 46.6 | 43.0 | 42.6 | 47.9 | 50.4 | 48.3 | 47.0 | 47.4 | **54.9±0.2** |
| hopper-m | 56.2 | 59.4 | 51.8 | 67.6 | 59.8 | 87.5 | 59.3 | 53.0 | 66.2 | **100.6±1.9** |
| walker2d-m | 71.0 | 71.8 | -0.2 | 74.0 | 83.1 | 84.8 | 83.7 | 73.3 | 78.3 | **92.4±2.7** |
| halfcheetah-m-r | 36.4 | 42.2 | 36.3 | 36.6 | 44.8 | 42.7 | 44.6 | 45.5 | 44.2 | **51.4±0.3** |
| hopper-m-r | 21.8 | 60.9 | 52.2 | 82.7 | 69.8 | 98.5 | 60.9 | 88.7 | 94.7 | **101.9±1.4** |
| walker2d-m-r | 24.9 | 57.0 | 7.0 | 66.6 | 78.1 | 61.7 | 81.8 | 81.8 | 73.8 | **89.7±5.0** |
| halfcheetah-m-e | 59.6 | **95.4** | 46.0 | 86.8 | 64.9 | 75.1 | 90.7 | 75.6 | 86.7 | 91.1±4.2 |
| hopper-m-e | 51.7 | 106.9 | 50.6 | 107.6 | 100.1 | 108.6 | 98.0 | 105.6 | 91.5 | **110.4±3.4** |
| walker2d-m-e | 101.2 | 107.7 | 22.1 | 108.1 | 110.0 | 111.3 | 110.1 | 107.9 | 109.6 | **114.4±0.7** |
| halfcheetah-e | 92.9 | 89.9 | 92.7 | 87.7 | 81.7 | 88.2 | **96.7** | 96.3 | 95.0 | 95.9±0.3 |
| hopper-e | **110.9** | 109.0 | 54.6 | 94.2 | **109.5** | 106.9 | 107.8 | 96.5 | 109.4 | 111.5±2.2 |
| walker2d-e | 107.7 | 106.3 | 106.6 | 108.3 | 110.1 | 110.7 | 110.2 | 108.5 | 109.9 | **114.7±0.4** |
| halfcheetah-r | 2.6 | 2.2 | 2.3 | 2.2 | 6.1 | 2.3 | 11.0 | 17.5 | 13.1 | **28.8±1.3** |
| hopper-r | 4.1 | 7.8 | 3.9 | 5.4 | 9.2 | 5.6 | 8.5 | 7.9 | 7.9 | **20.4±10.4** |
| walker2d-r | 1.2 | 4.9 | **12.8** | 2.2 | 0.2 | 6.9 | 1.6 | 5.1 | 5.4 | 4.8±2.2 |
| locomotion total | 784.2 | 968.0 | 581.7 | 972.6 | 975.3 | 1041.2 | 1013.2 | 1010.2 | 1033.1 | **1182.8** |
| antmaze-u | 66.8 | 78.9 | 73.0 | 54.2 | 80.0 | 54.0 | 73.0 | 82.6 | 89.6 | **92.4±1.8** |
| antmaze-u-d | 56.8 | 55.0 | 61.0 | 41.2 | 52.0 | 57.8 | 47.0 | 10.2 | 65.6 | **75.4±8.1** |
| antmaze-m-p | 0.0 | 0.0 | 0.0 | 0.0 | 0.0 | 0.0 | 0.0 | 59.0 | 76.4 | **80.2±5.1** |
| antmaze-m-d | 0.0 | 0.0 | 8.0 | 0.0 | 0.2 | 0.6 | 0.2 | 46.6 | 72.8 | **77.2±6.1** |
| antmaze-l-p | 0.0 | 6.7 | 0.0 | 0.0 | 0.0 | 0.0 | 0.0 | 16.4 | 42.0 | **55.4±6.2** |
| antmaze-l-d | 0.0 | 2.2 | 0.0 | 0.0 | 0.0 | 0.2 | 0.0 | 3.2 | 46.0 | **58.8±4.5** |
| antmaze total | 123.6 | 142.8 | 142.0 | 95.4 | 132.2 | 112.6 | 120.2 | 218.0 | 392.4 | **439.4** |

and CQL [39]. For the latter, we compare to BC [58], OneStep RL [7], IQL [37], $\mathcal{X}$QL [21], and SQL [88]. We also include the sequence-modeling method Decision Transformer (DT) [8].

**Comparison with baselines.**  Aggregated results are displayed in Table 2. On the Gym locomotion tasks, DMG outperforms prior methods on most tasks and achieves the highest total score. On the much more challenging AntMaze tasks, DMG outperforms all the baselines by a large margin, especially in the most difficult large mazes. For detailed learning curves, please refer to Appendix D.3. According to [56], we also report the results of DMG over more random seeds in Appendix D.2.

**Runtime.**  We test the runtime of DMG and other baselines on a GeForce RTX 3090. As shown in Appendix D.1, the runtime of DMG is comparable to that of the fastest offline RL algorithm TD3BC.

## 5.2  Performance Improvement over In-sample Learning Approaches

DMG can be combined with various in-sample learning approaches. Besides IQL [37], we also apply DMG to two recent state-of-the-art in-sample algorithms, $\mathcal{X}$QL [21] and SQL [88]. As shown in Table 3 (and Table 2), DMG consistently and substantially improves upon these in-sample methods, particularly on sub-optimal datasets where generalization plays a crucial role in the pursuit of a better policy. This provides compelling empirical evidence that the performance of in-sample methods is largely confined by eschewing generalization beyond the dataset, while DMG effectively exploits generalization, achieving significantly improved performance across tasks.

Table 3: DMG combined with various in-sample approaches, showing averaged scores over 5 seeds.

| Dataset-v2 | $\mathcal{X}$QL (+DMG) | SQL(+DMG) |
|---|---|---|
| halfcheetah-m | 47.7 → **55.3** | 48.3 → **54.5** |
| hopper-m | 71.1 → **90.1** | 75.5 → **97.7** |
| walker2d-m | 81.5 → **88.7** | 84.2 → **89.8** |
| halfcheetah-m-r | 44.8 → **51.1** | 44.8 → **51.8** |
| hopper-m-r | 97.3 → **102.5** | 101.7 → **101.8** |
| walker2d-m-r | 75.9 → **90.0** | 77.2 → **95.2** |
| halfcheetah-m-e | 89.8 → **92.5** | **94.0** → 93.5 |
| hopper-m-e | 107.1 → **111.1** | 111.8 → 110.4 |
| walker2d-m-e | 110.1 → **111.3** | 110.0 → 109.6 |
| total | 725.3 → **792.7** | 747.5 → **804.2** |

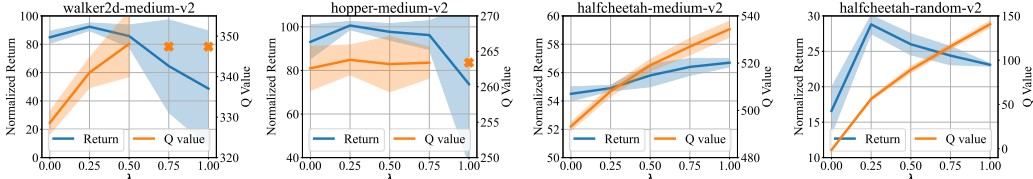

Figure 1: Performance and Q values of DMG with varying mixture coefficient $\lambda$ over 5 random seeds. The crosses $\times$ mean that the value functions diverge in several seeds. As $\lambda$ increases, DMG enables stronger generalization propagation, resulting in higher and probably divergent learned Q values. Mild generalization propagation plays a crucial role in achieving strong performance.

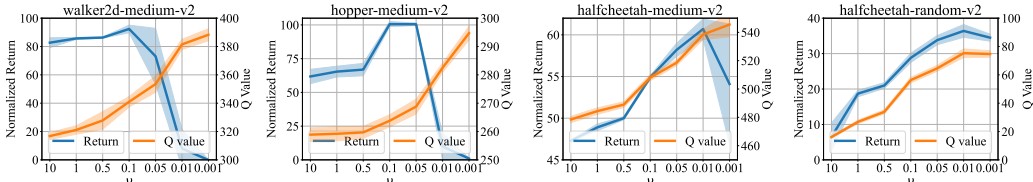

Figure 2: Performance and Q values of DMG with varying penalty coefficient $\nu$ over 5 random seeds. As $\nu$ decreases, DMG allows broader action generalization, leading to larger learned Q values. Mild action generalization is also critical for attaining superior performance.

### 5.3 Ablation Study for Performance and Value Estimation

**Mixture coefficient $\lambda$.** The mixture coefficient $\lambda$ controls the extent of generalization propagation. We fix $\nu = 0.1$ and vary $\lambda \in [0, 1]$, presenting the learned Q values and performance on several tasks in Figure 1. As $\lambda$ increases, DMG enables increased generalization propagation through bootstrapping, and the learned Q values become larger and probably diverge. A moderate $\lambda$ (mild generalization propagation) is crucial for achieving strong performance across datasets. Under the same degree of action generalization, mild generalization propagation effectively suppresses value overestimation, facilitating more stable policy learning.

**Penalty coefficient $\nu$.** The penalty coefficient $\nu$ regulates the degree of action generalization. We fix $\lambda = 0.25$ and vary $\nu$. The results are shown in Figure 2. As $\nu$ decreases, DMG allows broader action generalization beyond the dataset, which results in higher learned values. Regarding performance, a moderate $\nu$ (mild action generalization) is also crucial for achieving superior performance.

### 5.4 Online Fine-tuning after Offline RL

Benefiting from its flexibility in both generalization aspects, DMG enjoys a seamless transition from offline to online learning. This is accomplished through a gradual enhancement of both action generalization and generalization propagation. Since IQL [37] has demonstrated superior online fine-tuning performance compared to previous methods [55, 39] in its paper, we follow the experimental setup of IQL and compare to IQL. We also train online RL algorithm TD3 [18] from scratch for comparison. We use the challenging AntMaze domains [16], given DMG's already high offline performance in Gym locomotion domains.

Table 4: Online fine-tuning results on AntMaze tasks, showing normalized scores of offline training and 1M steps online fine-tuning, averaged over 5 seeds.

| Dataset-v2 | TD3 | IQL | DMG (Ours) |
|---|---|---|---|
| antmaze-u | 0.0 | $89.6 \rightarrow 96.2$ | $92.4 \rightarrow \mathbf{98.4}$ |
| antmaze-u-d | 0.0 | $65.6 \rightarrow 62.2$ | $75.4 \rightarrow \mathbf{89.2}$ |
| antmaze-m-p | 0.0 | $76.4 \rightarrow 89.8$ | $80.2 \rightarrow \mathbf{96.8}$ |
| antmaze-m-d | 0.0 | $72.8 \rightarrow 90.2$ | $77.2 \rightarrow \mathbf{96.2}$ |
| antmaze-l-p | 0.0 | $42.0 \rightarrow 78.6$ | $55.4 \rightarrow \mathbf{86.8}$ |
| antmaze-l-d | 0.0 | $46.0 \rightarrow 73.4$ | $58.8 \rightarrow \mathbf{89.0}$ |
| antmaze total | 0.0 | $392.4 \rightarrow 490.4$ | $439.4 \rightarrow \mathbf{556.4}$ |

mance in Gym locomotion domains. Results are presented in Table 4. While online training from scratch fails in the challenging sparse reward AntMaze tasks, DMG initialized with offline pretraining succeeds in learning near-optimal policies, outperforming IQL by a significant margin. Please refer to Appendix C.2 for experimental details, and to Appendix D.4 for learning curves.

# 6 Conclusion and Limitations

This work scrutinizes offline RL through the lens of generalization and proposes DMG, comprising mild action generalization and mild generalization propagation, to exploit generalization in offline RL appropriately. We theoretically analyze DMG in oracle and worst-case generalization scenarios, and empirically demonstrate its SOTA performance in offline training and online fine-tuning experiments.

While our work contributes valuable insights, it also has limitations. The DMG principle is shown to be effective across most scenarios. However, when the function approximator employed is highly compatible with a specific task setting, the learned value functions may generalize well in the entire action space. In such case, DMG may underperform full generalization methods due to conservatism.

## Acknowledgment

We thank the anonymous reviewers for feedback on an early version of this paper. This work was supported by the National Key R&D Program of China under Grant 2018AAA0102801, National Natural Science Foundation of China under Grant 61827804.

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

# A   Extended Related Work

**Model-free offline RL.**    In offline RL, a fixed dataset is provided and no further interactions are allowed [40, 42]. As a result, conventional off-policy RL algorithms suffer from the extrapolation error due to OOD actions and exhibit poor performance [19]. To address this challenge, various offline RL algorithms have been developed, primarily categorized into model-free and model-based approaches. In model-free solutions, value regularization methods introduce conservatism in value estimation through direct penalization [39, 36, 46, 86, 11, 64, 51], or via value ensembles [2, 3, 89]. Policy constraint approaches enforce proximity between the trained policy and the behavior policy, either explicitly via divergence penalties [84, 38, 30, 17, 83], implicitly by weighted behavior cloning [9, 57, 55, 80, 49], or directly through specific parameterization of the policy [19, 23, 93]. Some recent efforts focus on learning the optimal policy within the dataset's support (known as in-support or in-sample optimal policy) in a theoretically sound manner [49, 51, 83]. These approaches are less influenced by the the dataset's average quality. Another popular branch of algorithms opts for in-sample learning, which formulates the Bellman target without querying the values of any unseen actions [7, 45, 37, 85, 92, 88, 21]. Among these, OneStep RL [7] evaluates the behavior policy via SARSA [70] and performs only one step of constrained policy improvement without off-policy evaluation. IQL [37] modifies the SARSA update, using expectile regression to approximate an upper expectile of the value distribution and enables multi-step dynamic programming. Following IQL, several recent works such as InAC [85], IAC [92], $\mathcal{X}$QL [21], and SQL [88] have developed different in-sample learning frameworks, further enhancing the performance of in-sample learning approaches. However, this work shows that the performance of in-sample approaches is confined by eschewing generalization beyond the offline dataset. In contrast, the proposed approach DMG utilizes doubly mild generalization to appropriately exploit generalization and achieves significantly stronger performance across tasks.

**Model-based offline RL.**    Model-based offline RL methods involve training an environmental dynamics model, from which synthetic data is generated to facilitate policy optimization [69, 29, 32]. In the context of offline RL, algorithms such as MOPO [90] and MOReL [33] propose to estimate the uncertainty within the trained model and subsequently impose penalties or constraints on state-action pairs characterized by high uncertainty levels, thus achieving conservatism in the learning process. Some model-based approaches incorporate conservatism in a similar way to those model-free ones. For example, COMBO [91] leverages value penalization, while BREMEN [52] employs behavior regularization. More recently, MOBILE [68] introduces uncertainty quantification via the inconsistency of Bellman estimations within a learned dynamics ensemble. SCAS [50] proposes a generic model-based regularizer that unifies OOD state correction and OOD action suppression in offline RL. However, typical model-based methods often involve heavy computational overhead [29], and their effectiveness hinges on the accuracy of the trained dynamics model [54].

Recently, Bose et al. [5] explores multi-task offline RL from the perspective of representation learning and introduced a notion of neighborhood occupancy density. The neighborhood occupancy density at a given stata-action pair in the dataset for a source task is defined as the fraction of points in the dataset within a certain distance from that stata-action pair in the representation space. Bose et al. [5] use this concept to bound the representational transfer error in the downstream target task. In contrast, DMG is a wildly compatible idea in offline RL and provides insights into many offline RL methods. DMG balances the need for generalization with the risk of over-generalization in offline RL. Generalization to stata-action pairs in the neighborhood of the dataset corresponds to mild action generalization in the DMG framework.

# B   Proofs

In this section, we provide the proofs of all the theories in the paper.

## B.1   Proof of Theorem 1

This section presents the formal theorem for the Theorem 1 in the main paper, along with its proof.

We first make several common continuity assumptions for Theorem 1.

**Assumption 4** (Lipschitz $Q$). *The learned value function $Q_\theta$ is $K_Q$-Lipschitz and is upper bounded by $Q_{\max}$. $\forall s \sim \mathcal{D}, \forall a_1, a_2 \sim \mathcal{A}, |Q_\theta(s,a_1) - Q_\theta(s,a_2)| \le K_Q \|a_1 - a_2\|$.*

**Assumption 5** (Lipschitz $Q$ gradient). *The learned value function $Q_\theta$ is smooth, i.e, has a $K_g$-Lipschitz continuous gradient. $\forall s \sim \mathcal{D}, \forall a_1, a_2 \sim \mathcal{A}, \|\nabla_\theta Q_\theta(s,a_1) - \nabla_\theta Q_\theta(s,a_2)\| \le K_g \|a_1 - a_2\|$.*

**Assumption 6** (Bounded $Q$ and $Q$ gradient). *$\forall s, a, |Q_\theta(s,a)| \le Q_{\max}$ and $\|\nabla_\theta Q_\theta(s,a)\| \le g_{\max}$.*

**Assumption 7** (Lipschitz $P$). *The transition dynamics $P$ is $K_P$-Lipschitz. $\forall s, s' \sim \mathcal{S}, \forall a_1, a_2 \sim \mathcal{A}, |P(s'|s,a_1) - P(s'|s,a_2)| \le K_P \|a_1 - a_2\|$.*

**Assumption 8** (Lipschitz $R$). *The reward function $R$ is $K_R$-Lipschitz. $\forall s \sim \mathcal{S}, \forall a_1, a_2 \sim \mathcal{A}, |R(s|a_1) - R(s,a_2)| \le K_R \|a_1 - a_2\|$.*

A continuous learned Q function is particularly necessary for the analysis of value function generalization. Since we often use neural networks or linear models to parameterize the value function $Q_\theta$, Assumptions 4 and 5 can be relatively easily satisfied [24]. Assumptions 6, 7, and 8 are also common in the theoretical studies of RL [13, 87, 61] and optimization [6].

Before we start the proof of Theorem 1, we prove the following lemma.

**Lemma 2.** *$\forall s \sim \mathcal{D}, \forall a_1, a_2 \sim \mathcal{A}, |\mathcal{T}_u Q_\theta(s,a_1) - \mathcal{T}_u Q_\theta(s,a_2)| \le K_\mathcal{T} \|a_1 - a_2\|$. where $K_\mathcal{T}$ is a positive bounded constant.*

*Proof.* $\forall s \sim \mathcal{D}, \forall a_1, a_2 \sim \mathcal{A}$,

$$|\mathcal{T}_u Q_\theta(s,a_1) - \mathcal{T}_u Q_\theta(s,a_2)|$$

$$= \left| R(s,a_1) - R(s,a_2) + \gamma \mathbb{E}_{s' \sim P(\cdot|s,a_1)} \left[ \max_{a' \sim u(\cdot|s')} Q(s',a') \right] - \gamma \mathbb{E}_{s' \sim P(\cdot|s,a_2)} \left[ \max_{a' \sim u(\cdot|s')} Q(s',a') \right] \right|$$

$$\le |R(s,a_1) - R(s,a_2)| + \gamma \left| \mathbb{E}_{s' \sim P(\cdot|s,a_1)} \left[ \max_{a' \sim u(\cdot|s')} Q(s',a') \right] - \mathbb{E}_{s' \sim P(\cdot|s,a_2)} \left[ \max_{a' \sim u(\cdot|s')} Q(s',a') \right] \right|$$

$$= |R(s,a_1) - R(s,a_2)| + \gamma \left| \sum_{s'} (P(s'|s,a_1) - P(s'|s,a_2)) \max_{a' \sim u(\cdot|s')} Q(s',a') \right|$$

$$\le |R(s,a_1) - R(s,a_2)| + \gamma \sum_{s'} |(P(s'|s,a_1) - P(s'|s,a_2))| \left| \max_{a' \sim u(\cdot|s')} Q(s',a') \right|$$

$$\le K_R \|a_1 - a_2\| + \gamma \sum_{s'} K_P \|a_1 - a_2\| Q_{\max}$$

$$= (K_R + \gamma K_P |\mathcal{S}| Q_{\max}) \|a_1 - a_2\|$$

where the last inequality holds by Assumptions 6, 7, and 8.

Therefore, for any $s \sim \mathcal{D}, a_1, a_2 \sim \mathcal{A}$, it holds that

$$|\mathcal{T}_u Q_\theta(s,a_1) - \mathcal{T}_u Q_\theta(s,a_2)| \le K_\mathcal{T} \|a_1 - a_2\|, \tag{17}$$

where $K_\mathcal{T} := K_R + \gamma K_P |\mathcal{S}| Q_{\max}$ is a positive bounded constant. $\square$

We restate the scenario analyzed in Theorem 1: $Q_\theta$ is updated to $Q_{\theta'}$ by one gradient step on a single state-action pair $(s,a) \in \mathcal{D}$, which affects the Q-value of an arbitrary state-action pair $(s,\tilde{a}) \notin \mathcal{D}$. The parameter update is

$$\theta' = \theta + \alpha(\mathcal{T}_u Q_\theta(s,a) - Q_\theta(s,a))\nabla_\theta Q_\theta(s,a) \tag{18}$$

where $\alpha$ is the learning rate.

Now we start the proof of Theorem 1 in the main paper.

**Theorem 6** (Theorem 1). *Under Assumptions 4 to 8, the following equation holds when the learning rate $\alpha$ is sufficiently small and $\tilde{a}$ is sufficiently close to $a$:*

$$Q_{\theta'}(s,\tilde{a}) = Q_\theta(s,\tilde{a}) + C_1 \left( \mathcal{T}_u Q_\theta(s,\tilde{a}) - Q_\theta(s,\tilde{a}) + C_2 \|\tilde{a} - a\| \right) + \mathcal{O} \left( \|\theta' - \theta\|^2 \right) \tag{19}$$

*where $C_1 \in [0,1]$ and $C_2 \in [-K_Q - K_R - \gamma K_P |\mathcal{S}| Q_{\max}, K_Q + K_R + \gamma K_P |\mathcal{S}| Q_{\max}]$.*

*Proof.* We formalize $Q_{\theta'}(s, \tilde{a})$ by Taylor expansion at the parameter $\theta$:

$$Q_{\theta'}(s, \tilde{a}) = Q_\theta(s, \tilde{a}) + \nabla_\theta Q_\theta(s, \tilde{a})^\top (\theta' - \theta) + \mathcal{O}\left(\|\theta' - \theta\|^2\right) \tag{20}$$

By plugging Eq. (18) into Eq. (20), we have

$$Q_{\theta'}(s, \tilde{a}) = Q_\theta(s, \tilde{a}) + \alpha \nabla_\theta Q_\theta(s, \tilde{a})^\top \nabla_\theta Q_\theta(s, a) \left(\mathcal{T}_u Q_\theta(s, a) - Q_\theta(s, a)\right) + \mathcal{O}\left(\|\theta' - \theta\|^2\right) \tag{21}$$

According to Assumption 4 and Lemma 2, it holds that

$$|Q_\theta(s, \tilde{a}) - Q_\theta(s, a)| \leq K_Q \|\tilde{a} - a\| \tag{22}$$

$$|\mathcal{T}_u Q_\theta(s, \tilde{a}) - \mathcal{T}_u Q_\theta(s, a)| \leq K_\mathcal{T} \|\tilde{a} - a\| \tag{23}$$

where $K_\mathcal{T} := K_R + \gamma K_P |\mathcal{S}| Q_{\max}$ is a positive bounded constant.

Therefore,

$$\begin{aligned}
&|(\mathcal{T}_u Q_\theta(s, \tilde{a}) - Q_\theta(s, \tilde{a})) - (\mathcal{T}_u Q_\theta(s, a) - Q_\theta(s, a))| \\
=&|(\mathcal{T}_u Q_\theta(s, \tilde{a}) - \mathcal{T}_u Q_\theta(s, a)) + (Q_\theta(s, a) - Q_\theta(s, \tilde{a}))| \\
\leq&|(\mathcal{T}_u Q_\theta(s, \tilde{a}) - \mathcal{T}_u Q_\theta(s, a))| + |(Q_\theta(s, a) - Q_\theta(s, \tilde{a}))| \\
\leq& K_\mathcal{T} \|\tilde{a} - a\| + K_Q \|\tilde{a} - a\|
\end{aligned}$$

As a result,

$$\mathcal{T}_u Q_\theta(s, a) - Q_\theta(s, a) \leq \mathcal{T}_u Q_\theta(s, \tilde{a}) - Q_\theta(s, \tilde{a}) + (K_Q + K_\mathcal{T}) \|\tilde{a} - a\|$$

$$\mathcal{T}_u Q_\theta(s, a) - Q_\theta(s, a) \geq \mathcal{T}_u Q_\theta(s, \tilde{a}) - Q_\theta(s, \tilde{a}) - (K_Q + K_\mathcal{T}) \|\tilde{a} - a\|$$

Thus we can let

$$\mathcal{T}_u Q_\theta(s, a) - Q_\theta(s, a) = \mathcal{T}_u Q_\theta(s, \tilde{a}) - Q_\theta(s, \tilde{a}) + C_2 \|\tilde{a} - a\|, \tag{24}$$

where $C_2 \in [-K_Q - K_\mathcal{T}, K_Q + K_\mathcal{T}]$ is a bounded constant.

Now we shift our focus to $\alpha \nabla_\theta Q_\theta(s, \tilde{a})^\top \nabla_\theta Q_\theta(s, a)$. Let $v = \nabla_\theta Q_\theta(s, \tilde{a}) - \nabla_\theta Q_\theta(s, a)$. According to the smoothness of $Q_\theta$ in Assumption 5, it holds that

$$\|v\| = \|\nabla_\theta Q_\theta(s, \tilde{a}) - \nabla_\theta Q_\theta(s, a)\| \leq K_g \|\tilde{a} - a\|. \tag{25}$$

Therefore,

$$\begin{aligned}
&\nabla_\theta Q_\theta(s, \tilde{a})^\top \nabla_\theta Q_\theta(s, a) \\
=&(\nabla_\theta Q_\theta(s, a) + v)^\top \nabla_\theta Q_\theta(s, a) \\
=&\|\nabla_\theta Q_\theta(s, a)\|^2 + v^\top \nabla_\theta Q_\theta(s, a) \\
\geq&\|\nabla_\theta Q_\theta(s, a)\|^2 - \|v\|\|\nabla_\theta Q_\theta(s, a)\| \\
\geq&\|\nabla_\theta Q_\theta(s, a)\|^2 - K_g \|\tilde{a} - a\|\|\nabla_\theta Q_\theta(s, a)\|
\end{aligned}$$

Therefore, for sufficiently close $\tilde{a}$ and $a$ such that $\|\tilde{a} - a\| \leq \|\nabla_\theta Q_\theta(s, a)\|/K_g$, it holds that $\alpha \nabla_\theta Q_\theta(s, \tilde{a})^\top \nabla_\theta Q_\theta(s, a) \geq 0$.

On the other hand, because $\|\nabla_\theta Q_\theta\|$ is bounded by $g_{\max}$ according to Assumption 6, it holds that

$$\alpha \nabla_\theta Q_\theta(s, \tilde{a})^\top \nabla_\theta Q_\theta(s, a) \leq \alpha g_{\max}^2$$

By choosing a small learning rate $\alpha$ such that $\alpha \leq 1/g_{\max}^2$,

$$\alpha \nabla_\theta Q_\theta(s, \tilde{a})^\top \nabla_\theta Q_\theta(s, a) \leq 1$$

In such cases (sufficiently close $\tilde{a}$ and $a$, and sufficiently small $\alpha$), let

$$C_1 := \alpha \nabla_\theta Q_\theta(s, \tilde{a})^\top \nabla_\theta Q_\theta(s, a) \tag{26}$$

We have $C_1 \in [0, 1]$.

By plugging Equations (24) and (26) into Equation (21), the following equation holds.

$$Q_{\theta'}(s, \tilde{a}) = Q_\theta(s, \tilde{a}) + C_1 \left(\mathcal{T}_u Q_\theta(s, \tilde{a}) - Q_\theta(s, \tilde{a}) + C_2 \|\tilde{a} - a\|\right) + \mathcal{O}\left(\|\theta' - \theta\|^2\right) \tag{27}$$

where $C_1 \in [0, 1]$, $C_2 \in [-K_Q - K_\mathcal{T}, K_Q + K_\mathcal{T}]$, and $K_\mathcal{T} = K_R + \gamma K_P |\mathcal{S}| Q_{\max}$.

This concludes the proof.

$\square$

## B.2 Proofs under Oracle Generalization

We first restate the several definitions in the main paper.

**Definition 4** (Mildly generalized policy, Definition 1). *Policy $\tilde{\beta}$ is termed a mildly generalized policy if it satisfies*

$$\text{supp}(\hat{\beta}(\cdot|s)) \subseteq \text{supp}(\tilde{\beta}(\cdot|s)), \quad and \quad \max_{a_1 \sim \tilde{\beta}(\cdot|s)} \min_{a_2 \sim \hat{\beta}(\cdot|s)} \|a_1 - a_2\| \le \epsilon_a, \tag{28}$$

*where $\hat{\beta}$ is the empirical behavior policy in the offline dataset.*

**Definition 5** (Definition 2). *The Doubly Mildly Generalization (DMG) operator is defined as*

$$\mathcal{T}_{\text{DMG}}Q(s,a) := R(s,a) + \gamma\mathbb{E}_{s' \sim P(\cdot|s,a)}\left[\lambda \max_{a' \sim \tilde{\beta}(\cdot|s')} Q(s',a') + (1-\lambda) \max_{a' \sim \hat{\beta}(\cdot|s')} Q(s',a')\right] \tag{29}$$

*where $\hat{\beta}$ is the empirical behavior policy in the dataset and $\tilde{\beta}$ is a mildly generalized policy.*

**Definition 6** (Definition 3). *The In-sample Q Learning operator [37] is defined as*

$$\mathcal{T}_{\text{In}}Q(s,a) := R(s,a) + \gamma\mathbb{E}_{s' \sim P(\cdot|s,a)}\left[\max_{a' \sim \hat{\beta}(\cdot|s')} Q(s',a')\right] \tag{30}$$

*where $\hat{\beta}$ is the empirical behavior policy in the dataset.*

In this subsection, we assume that the learned value function can make oracle generalization in the mild generalization area $\tilde{\beta}(a|s) > 0$, which is formally defined as follows.

**Assumption 9** (Oracle generalization, Assumption 1). *The generalization of learned Q functions in the mild generalization area $\tilde{\beta}(a|s) > 0$ reflects the true value updates according to $\mathcal{T}_{\text{DMG}}$. In other words, $\mathcal{T}_{\text{DMG}}$ is well defined in the mild generalization area $\tilde{\beta}(a|s) > 0$.*

This assumption can be considered reasonable according to the results presented in Theorem 6 above. In such cases, we can analyze the dynamic programming properties of operators $\mathcal{T}_{\text{In}}$ and $\mathcal{T}_{\text{DMG}}$.

Before we start the proofs of Lemma 1 and Theorem 2 in the main paper, we prove a lemma.

**Lemma 3.** *For any function $f_1$, $f_2$, any variant $x \in \mathcal{X}$, the following inequality holds:*

$$\left|\max_{x \in \mathcal{X}} f_1(x) - \max_{x \in \mathcal{X}} f_2(x)\right| \le \max_{x \in \mathcal{X}} |f_1(x) - f_2(x)|. \tag{31}$$

*Proof.* Define $x_1 := \text{argmax}_{x \in \mathcal{X}} f_1(x)$ and $x_2 := \text{argmax}_{x \in \mathcal{X}} f_2(x)$.

According to the definition, the following inequality holds:

$$f_1(x_2) - f_2(x_2) \le f_1(x_1) - f_2(x_2) \le f_1(x_1) - f_2(x_1) \tag{32}$$

Therefore,

$$\left|\max_{x \in \mathcal{X}} f_1(x) - \max_{x \in \mathcal{X}} f_2(x)\right|$$
$$= |f_1(x_1) - f_2(x_2)|$$
$$\le \max\{|f_1(x_2) - f_2(x_2)|, |f_1(x_1) - f_2(x_1)|\}$$
$$\le \max_{x \in \mathcal{X}} |f_1(x) - f_2(x)|$$

This concludes the proof of Lemma 3. $\square$

**Lemma 4** (Lemma 1). *$\mathcal{T}_{\text{In}}$ is a $\gamma$-contraction operator in the in-sample area $\hat{\beta}(a|s) > 0$ under the $\mathcal{L}_\infty$ norm.*

*Proof.* Let $f_1$ and $f_2$ be two arbitrary functions.

For all $(s,a)$ $s.t.$ $\hat{\beta}(a|s) > 0$, we have

$$|\mathcal{T}_{\text{In}}f_1(s,a) - \mathcal{T}_{\text{In}}f_2(s,a)|$$

$$= \left| R(s,a) + \gamma \mathbb{E}_{s'\sim P(\cdot|s,a)} \left[ \max_{a'\sim\hat{\beta}(\cdot|s')} f_1(s',a') \right] - R(s,a) - \gamma \mathbb{E}_{s'\sim P(\cdot|s,a)} \left[ \max_{a'\sim\hat{\beta}(\cdot|s')} f_2(s',a') \right] \right|$$

$$= \gamma \left| \mathbb{E}_{s'\sim P(\cdot|s,a)} \left[ \max_{a'\sim\hat{\beta}(\cdot|s')} f_1(s',a') - \max_{a'\sim\hat{\beta}(\cdot|s')} f_2(s',a') \right] \right|$$

$$\leq \gamma \mathbb{E}_{s'\sim P(\cdot|s,a)} \left[ \left| \max_{a'\sim\hat{\beta}(\cdot|s')} f_1(s',a') - \max_{a'\sim\hat{\beta}(\cdot|s')} f_2(s',a') \right| \right]$$

$$\leq \gamma \mathbb{E}_{s'\sim P(\cdot|s,a)} \left[ \max_{a'\sim\hat{\beta}(\cdot|s')} |f_1(s',a') - f_2(s',a')| \right]$$

$$\leq \gamma \max_{(s,a):\hat{\beta}(a|s)>0} |f_1(s,a) - f_2(s,a)|$$

where the second inequality holds by Lemma 3.

Therefore, in the in-sample area $\tilde{\beta}(a|s) > 0$, $\mathcal{T}_{\text{In}}$ is a $\gamma$-contraction operator under the $\mathcal{L}_\infty$ norm. This concludes the proof for $\mathcal{T}_{\text{In}}$. $\qquad\square$

Thus, by repeatedly applying $\mathcal{T}_{\text{In}}$, any initial Q function can converge to the unique fixed point $Q_{\text{In}}^*$. We denote its induced policy by $\pi_{\text{In}}^*$:

$$Q_{\text{In}}^*(s,a) = R(s,a) + \gamma \mathbb{E}_{s'\sim P(\cdot|s,a)} \left[ \max_{a'\sim\hat{\beta}(\cdot|s')} Q_{\text{In}}^*(s',a') \right], \quad \hat{\beta}(a|s) > 0, \tag{33}$$

$$\pi_{\text{In}}^*(s) := \underset{a\sim\hat{\beta}(\cdot|s)}{\text{argmax}}\, Q_{\text{In}}^*(s,a). \tag{34}$$

Here, $Q_{\text{In}}^*$ is known as the in-sample optimal value function [38, 37], which is the value function of the in-sample optimal policy $\pi_{\text{In}}^*$. We refer readers to [83, 37, 49, 51] for more discussions on the in-sample or in-support optimality.

Now we start the proof of Theorem 2 in the main paper.

**Theorem 7** (Contraction, Theorem 2). *Under Assumption 9, $\mathcal{T}_{\text{DMG}}$ is a $\gamma$-contraction operator in the mild generalization area $\tilde{\beta}(a|s) > 0$ under the $\mathcal{L}_\infty$ norm. Therefore, by repeatedly applying $\mathcal{T}_{\text{DMG}}$, any initial Q function can converge to the unique fixed point $Q_{\text{DMG}}^*$.*

*Proof.* By the oracle generalization assumption (Assumption 9), $\mathcal{T}_{\text{DMG}}$ is well defined in the mild generalization area $\tilde{\beta}(a|s) > 0$.

Let $f_1$ and $f_2$ be two arbitrary functions. For all $(s,a)$ $s.t.$ $\tilde{\beta}(a|s) > 0$, we have

$$\mathcal{T}_{\text{DMG}}f_1(s,a) - \mathcal{T}_{\text{DMG}}f_2(s,a)$$

$$= R(s,a) + \gamma \mathbb{E}_{s'\sim P(\cdot|s,a)} \left[ \lambda \max_{a'\sim\tilde{\beta}(\cdot|s')} f_1(s',a') + (1-\lambda) \max_{a'\sim\hat{\beta}(\cdot|s')} f_1(s',a') \right]$$

$$\quad - R(s,a) - \gamma \mathbb{E}_{s'\sim P(\cdot|s,a)} \left[ \lambda \max_{a'\sim\tilde{\beta}(\cdot|s')} f_2(s',a') + (1-\lambda) \max_{a'\sim\hat{\beta}(\cdot|s')} f_2(s',a') \right]$$

$$= \gamma\lambda \mathbb{E}_{s'\sim P(\cdot|s,a)} \left[ \max_{a'\sim\tilde{\beta}(\cdot|s')} f_1(s',a') - \max_{a'\sim\tilde{\beta}(\cdot|s')} f_2(s',a') \right]$$

$$\quad + \gamma(1-\lambda) \mathbb{E}_{s'\sim P(\cdot|s,a)} \left[ \max_{a'\sim\hat{\beta}(\cdot|s')} f_1(s',a') - \max_{a'\sim\hat{\beta}(\cdot|s')} f_2(s',a') \right]$$

Therefore, for all $(s, a)$ s.t. $\tilde{\beta}(a|s) > 0$,

$$|\mathcal{T}_{\mathrm{DMG}} f_1(s, a) - \mathcal{T}_{\mathrm{DMG}} f_2(s, a)|$$

$$\leq \left| \gamma\lambda\mathbb{E}_{s' \sim P(\cdot|s,a)} \left[ \max_{a' \sim \tilde{\beta}(\cdot|s')} f_1(s', a') - \max_{a' \sim \tilde{\beta}(\cdot|s')} f_2(s', a') \right] \right|$$

$$+ \left| \gamma(1 - \lambda)\mathbb{E}_{s' \sim P(\cdot|s,a)} \left[ \max_{a' \sim \hat{\beta}(\cdot|s')} f_1(s', a') - \max_{a' \sim \hat{\beta}(\cdot|s')} f_2(s', a') \right] \right|$$

$$\leq \gamma\lambda\mathbb{E}_{s' \sim P(\cdot|s,a)} \left[ \left| \max_{a' \sim \tilde{\beta}(\cdot|s')} f_1(s', a') - \max_{a' \sim \tilde{\beta}(\cdot|s')} f_2(s', a') \right| \right]$$

$$+ \gamma(1 - \lambda)\mathbb{E}_{s' \sim P(\cdot|s,a)} \left[ \left| \max_{a' \sim \hat{\beta}(\cdot|s')} f_1(s', a') - \max_{a' \sim \hat{\beta}(\cdot|s')} f_2(s', a') \right| \right]$$

$$\leq \gamma\lambda\mathbb{E}_{s' \sim P(\cdot|s,a)} \left[ \max_{a' \sim \tilde{\beta}(\cdot|s')} |f_1(s', a') - f_2(s', a')| \right]$$

$$+ \gamma(1 - \lambda)\mathbb{E}_{s' \sim P(\cdot|s,a)} \left[ \max_{a' \sim \hat{\beta}(\cdot|s')} |f_1(s', a') - f_2(s', a')| \right]$$

$$\leq \gamma\lambda\mathbb{E}_{s' \sim P(\cdot|s,a)} \max_{(s,a):\tilde{\beta}(a|s)>0} |f_1(s, a) - f_2(s, a)|$$

$$+ \gamma(1 - \lambda)\mathbb{E}_{s' \sim P(\cdot|s,a)} \max_{(s,a):\tilde{\beta}(a|s)>0} |f_1(s, a) - f_2(s, a)|$$

$$= \gamma \max_{(s,a):\tilde{\beta}(a|s)>0} |f_1(s, a) - f_2(s, a)|$$

where the third inequality holds by Lemma 3.

Therefore, in the mild generalization area $\tilde{\beta}(a|s) > 0$, $\mathcal{T}_{\mathrm{DMG}}$ is a $\gamma$-contraction operator under the $\mathcal{L}_\infty$ norm. This concludes the proof. $\qquad\square$

As a result, by repeatedly applying $\mathcal{T}_{\mathrm{DMG}}$, any initial Q function can converge to the unique fixed point $Q^*_{\mathrm{DMG}}$. We denote the induced policy of $Q^*_{\mathrm{DMG}}$ by $\pi^*_{\mathrm{DMG}}$.

$$Q^*_{\mathrm{DMG}}(s, a) = R(s, a) + \gamma\mathbb{E}_{s' \sim P(\cdot|s,a)} \left[ \max_{a' \sim \tilde{\beta}(\cdot|s')} Q^*_{\mathrm{DMG}}(s', a') \right], \quad \tilde{\beta}(a|s) > 0, \qquad (35)$$

$$\pi^*_{\mathrm{DMG}}(s) := \underset{a \sim \tilde{\beta}(\cdot|s)}{\operatorname{argmax}} Q^*_{\mathrm{DMG}}(s, a). \qquad (36)$$

Before we start the proof of Theorem 3, we prove two lemmas.

**Lemma 5.** *Under Assumption 9, for any function $f$, the following inequality holds:*

$$\mathcal{T}_{\mathrm{DMG}} f(s, a) \geq \mathcal{T}_{\mathrm{In}} f(s, a), \quad \forall (s, a) \text{ s.t. } \tilde{\beta}(a|s) > 0. \qquad (37)$$

*Proof.* The oracle generalization assumption (Assumption 9) implies that $\mathcal{T}_{\mathrm{In}}$ is also well defined in the mild generalization area $\tilde{\beta}(a|s) > 0$. Because $\mathrm{supp}(\hat{\beta}(\cdot|s)) \subseteq \mathrm{supp}(\tilde{\beta}(\cdot|s))$, $\mathcal{T}_{\mathrm{In}}$ requires less information than $\mathcal{T}_{\mathrm{DMG}}$. Therefore, $\mathcal{T}_{\mathrm{DMG}}$ being well defined in the mild generalization area implies $\mathcal{T}_{\mathrm{In}}$ also being well defined in that area.

According to the definitions, for all $(s, a)$ s.t. $\tilde{\beta}(a|s) > 0$,

$$\mathcal{T}_{\mathrm{DMG}} f(s, a) = R(s, a) + \gamma\mathbb{E}_{s' \sim P(\cdot|s,a)} \left[ \lambda \max_{a' \sim \tilde{\beta}(\cdot|s')} f(s', a') + (1 - \lambda) \max_{a' \sim \hat{\beta}(\cdot|s')} f(s', a') \right] \quad (38)$$

$$\mathcal{T}_{\mathrm{In}} f(s, a) = R(s, a) + \gamma\mathbb{E}_{s' \sim P(\cdot|s,a)} \left[ \max_{a' \sim \hat{\beta}(\cdot|s')} f(s', a') \right] \qquad (39)$$

Therefore, for all $(s, a)$ s.t. $\tilde{\beta}(a|s) > 0$, we have

$$\mathcal{T}_{\text{DMG}} f(s, a) - \mathcal{T}_{\text{In}} f(s, a)$$

$$= \gamma \mathbb{E}_{s' \sim P(\cdot|s,a)} \left[ \lambda \max_{a' \sim \tilde{\beta}(\cdot|s')} f(s', a') - \lambda \max_{a' \sim \hat{\beta}(\cdot|s')} f(s', a') \right]$$

$$\geq 0$$

where the last inequality holds because $\tilde{\beta}$ has a wider support than $\hat{\beta}$. $\qquad\square$

**Lemma 6.** *Under Assumption 9, for any function $f_1, f_2$ such that $f_1(s, a) \geq f_2(s, a)$, $\forall(s, a)$ s.t. $\tilde{\beta}(a|s) > 0$, the following inequality holds:*

$$\mathcal{T}_{\text{DMG}} f_1(s, a) \geq \mathcal{T}_{\text{DMG}} f_2(s, a), \quad \forall(s, a) \text{ s.t. } \tilde{\beta}(a|s) > 0 \tag{40}$$

*Proof.* By Assumption 9, $\mathcal{T}_{\text{DMG}}$ is well defined in the mild generalization area $\tilde{\beta}(a|s) > 0$.

According to the definition, for all $(s, a)$ s.t. $\tilde{\beta}(a|s) > 0$,

$$\mathcal{T}_{\text{DMG}} f(s, a) = R(s, a) + \gamma \mathbb{E}_{s' \sim P(\cdot|s,a)} \left[ \lambda \max_{a' \sim \tilde{\beta}(\cdot|s')} f(s', a') + (1 - \lambda) \max_{a' \sim \hat{\beta}(\cdot|s')} f(s', a') \right] \tag{41}$$

$f_1$ and $f_2$ satisfy

$$f_1(s, a) \geq f_2(s, a), \forall(s, a) \text{ s.t. } \tilde{\beta}(a|s) > 0. \tag{42}$$

Therefore, for all $(s, a)$ s.t. $\tilde{\beta}(a|s) > 0$,

$$\mathcal{T}_{\text{DMG}} f_1(s, a) - \mathcal{T}_{\text{DMG}} f_2(s, a)$$

$$= \gamma \mathbb{E}_{s' \sim P(\cdot|s,a)} \left[ \lambda \max_{a' \sim \tilde{\beta}(\cdot|s')} f_1(s', a') - \lambda \max_{a' \sim \tilde{\beta}(\cdot|s')} f_2(s', a') \right]$$

$$+ \gamma \mathbb{E}_{s' \sim P(\cdot|s,a)} \left[ (1 - \lambda) \max_{a' \sim \hat{\beta}(\cdot|s')} f_1(s', a') - (1 - \lambda) \max_{a' \sim \hat{\beta}(\cdot|s')} f_2(s', a') \right]$$

$$\geq 0$$

$\qquad\square$

Now we start the proof of Theorem 3 in the main paper.

**Theorem 8** (Performance, Theorem 3). *Under Assumption 9, the value functions of $\pi_{\text{DMG}}^*$ and $\pi_{\text{In}}^*$ satisfy:*

$$V^{\pi_{\text{DMG}}^*}(s) \geq V^{\pi_{\text{In}}^*}(s), \quad \forall s \in \mathcal{D}. \tag{43}$$

*Proof.* We first prove the following inequality:

$$(\mathcal{T}_{\text{DMG}})^k f(s, a) \geq (\mathcal{T}_{\text{In}})^k f(s, a), \ \forall k \in \mathbb{Z}^+, \ \forall f, \ \forall(s, a) \text{ s.t. } \tilde{\beta}(a|s) > 0. \tag{44}$$

When $k = 1$, according to Lemma 5, it holds that

$$(\mathcal{T}_{\text{DMG}})^1 f(s, a) \geq (\mathcal{T}_{\text{In}})^1 f(s, a), \ \forall f, \ \forall(s, a) \text{ s.t. } \tilde{\beta}(a|s) > 0.$$

Suppose when $k = i$, the following inequality holds:

$$(\mathcal{T}_{\text{DMG}})^i f(s, a) \geq (\mathcal{T}_{\text{In}})^i f(s, a), \ \forall f, \ \forall(s, a) \text{ s.t. } \tilde{\beta}(a|s) > 0.$$

Then $(\mathcal{T}_{\text{DMG}})^i f$ and $(\mathcal{T}_{\text{In}})^i f$ are the two functions $f_1, f_2$ that satisfy the condition in Lemma 6. Therefore, by Lemma 6, it holds that

$$\mathcal{T}_{\text{DMG}}(\mathcal{T}_{\text{DMG}})^i f(s, a) \geq \mathcal{T}_{\text{DMG}}(\mathcal{T}_{\text{In}})^i f(s, a), \ \forall f, \ \forall(s, a) \text{ s.t. } \tilde{\beta}(a|s) > 0. \tag{45}$$

Now considering $(\mathcal{T}_{\mathrm{In}})^i f$ as the function $f$ in Lemma 5. By Lemma 5, it holds that

$$\mathcal{T}_{\mathrm{DMG}}(\mathcal{T}_{\mathrm{In}})^i f(s,a) \geq \mathcal{T}_{\mathrm{In}}(\mathcal{T}_{\mathrm{In}})^i f(s,a), \; \forall f, \;\; \forall (s,a) \; s.t. \; \tilde{\beta}(a|s) > 0. \tag{46}$$

Combining Equations (45) and (46), we have

$$(\mathcal{T}_{\mathrm{DMG}})^{i+1} f(s,a) \geq (\mathcal{T}_{\mathrm{In}})^{i+1} f(s,a), \; \forall f, \;\; \forall (s,a) \; s.t. \; \tilde{\beta}(a|s) > 0.$$

Therefore, for all $k \in \mathbb{Z}^+$, the following inequality holds:

$$(\mathcal{T}_{\mathrm{DMG}})^k f(s,a) \geq (\mathcal{T}_{\mathrm{In}})^k f(s,a), \; \forall f, \;\; \forall (s,a) \; s.t. \; \tilde{\beta}(a|s) > 0. \tag{47}$$

Lemma 4 states that $\mathcal{T}_{\mathrm{In}}$ is a $\gamma$-contraction operator in the in-sample area $\hat{\beta}(a|s) > 0$. Thus we have

$$Q^*_{\mathrm{In}}(s,a) = \lim_{k \to \infty} (\mathcal{T}_{\mathrm{In}})^k f(s,a), \; \forall (s,a) \; s.t. \; \hat{\beta}(a|s) > 0. \tag{48}$$

Under Assumption 9, Theorem 7 states that $\mathcal{T}_{\mathrm{DMG}}$ is a $\gamma$-contraction operator in the mild generalization area $\tilde{\beta}(a|s) > 0$. Thus we have

$$Q^*_{\mathrm{DMG}}(s,a) = \lim_{k \to \infty} (\mathcal{T}_{\mathrm{DMG}})^k f(s,a), \; \forall (s,a) \; s.t. \; \tilde{\beta}(a|s) > 0. \tag{49}$$

As $\tilde{\beta}$ has a wider support than $\hat{\beta}$, $\mathrm{supp}(\hat{\beta}(\cdot|s)) \subseteq \mathrm{supp}(\tilde{\beta}(\cdot|s))$, the following inequality holds by combining Equations (47) to (49):

$$Q^*_{\mathrm{DMG}}(s,a) \geq Q^*_{\mathrm{In}}(s,a), \; \forall (s,a) \; s.t. \; \hat{\beta}(a|s) > 0. \tag{50}$$

Therefore, for any $s \sim \mathcal{D}$,

$$V^{\pi^*_{\mathrm{DMG}}}(s) = V^*_{\mathrm{DMG}}(s) = Q^*_{\mathrm{DMG}}(s, \pi^*_{\mathrm{DMG}}(s))$$
$$\geq Q^*_{\mathrm{DMG}}(s, \pi^*_{\mathrm{In}}(s))$$
$$\geq Q^*_{\mathrm{In}}(s, \pi^*_{\mathrm{In}}(s)) = V^*_{\mathrm{In}}(s) = V^{\pi^*_{\mathrm{In}}}(s)$$

where the first inequality holds because $\pi^*_{\mathrm{DMG}}(s) := \mathrm{argmax}_{a \sim \tilde{\beta}(\cdot|s)} Q^*_{\mathrm{DMG}}(s,a)$ and $\pi^*_{\mathrm{In}}(s) \in \hat{\beta}(\cdot|s)$ (thus $\pi^*_{\mathrm{In}}(s) \in \tilde{\beta}(\cdot|s)$), and the second inequality holds by Equation (50).

This concludes the proof. $\qquad\square$

Theorem 8 indicates that the policy induced by the DMG operator can behave better than the in-sample optimal policy under the oracle generalization condition.

### B.3 Proofs under Worst-case Generalization

In this section, we focus on the analyses in the worst-case generalization scenario, where the learned value functions may exhibit poor generalization in the mild generalization area $\tilde{\beta}(a|s) > 0$. In other words, this section considers that $\mathcal{T}_{\mathrm{DMG}}$ is only defined in the in-sample area $\hat{\beta}(a|s) > 0$ and the learned value functions may have any generalization error at other state-action pairs. In this case, we use the notation $\hat{\mathcal{T}}_{\mathrm{DMG}}$ for differentiation.

In this case, we make the following continuity assumptions about the learned $Q$ function and the transition dynamics $P$.

**Assumption 10** (Lipschitz Q). *The learned Q function is $K_Q$-Lipschitz. $\forall s \sim \mathcal{D}, \forall a_1, a_2 \sim \mathcal{A}$, $|Q(s,a_1) - Q(s,a_2)| \leq K_Q \|a_1 - a_2\|$*

**Assumption 11** (Lipschitz P). *The transition dynamics $P$ is $K_P$-Lipschitz. $\forall s, s' \sim \mathcal{S}, \forall a_1, a_2 \sim \mathcal{A}$, $|P(s'|s,a_1) - P(s'|s,a_2)| \leq K_P \|a_1 - a_2\|$*

For Assumption 10, a continuous learned Q function is particularly necessary for the analysis of value function generalization and can be relatively easily satisfied [24], since we often use neural networks or linear models to parameterize the value function. For Assumption 11, continuous transition dynamics is also a standard assumption in the theoretical studies of RL [13, 14, 87, 61]. Several previous works assume the transition to be Lipschitz continuous with respect to (w.r.t) both state and action [13, 14]. In our paper, we need the Lipschitz continuity to hold only w.r.t. action.

Before we start the proof of Theorem 4, we prove two lemmas.

**Lemma 7.** *Under Assumption 10, for any function $f$ and $s \sim \mathcal{D}$, the following inequality holds:*

$$\max_{a \sim \tilde{\beta}(\cdot|s)} f(s,a) - \max_{a \sim \hat{\beta}(\cdot|s)} f(s,a) \leq \epsilon_a K_Q. \tag{51}$$

*Proof.* For any $s \sim \mathcal{D}$, we define $\tilde{a}^*$, $\hat{a}^*$, $\hat{a}'$ as follows:

$$\tilde{a}^* = \underset{a \sim \tilde{\beta}(\cdot|s)}{\operatorname{argmax}} f(s,a) \tag{52}$$

$$\hat{a}^* = \underset{a \sim \hat{\beta}(\cdot|s)}{\operatorname{argmax}} f(s,a) \tag{53}$$

$$\hat{a}' = \underset{a \sim \hat{\beta}(\cdot|s)}{\operatorname{argmin}} \|\tilde{a}^* - a\| \tag{54}$$

According to the definition of mildly generalized policy $\tilde{\beta}$ (Definition 4), it holds that $\|\tilde{a}^* - \hat{a}'\| \leq \epsilon_a$. Further by Assumption 10, it holds that

$$|f(s, \tilde{a}^*) - f(s, \hat{a}')| \leq K_Q \|\tilde{a}^* - \hat{a}'\| \leq \epsilon_a K_Q, \ \ \forall s \sim \mathcal{D}.$$

Therefore,

$$f(s, \tilde{a}^*) - f(s, \hat{a}^*) \leq f(s, \tilde{a}^*) - f(s, \hat{a}') \leq \epsilon_a K_Q, \ \ \forall s \sim \mathcal{D}.$$

$\square$

**Lemma 8.** *For any function $f_1, f_2$ such that $f_1(s,a) \geq f_2(s,a)$, $\forall (s,a)$ s.t. $\hat{\beta}(a|s) > 0$, the following inequality holds:*

$$\mathcal{T}_{\text{In}} f_1(s,a) \geq \mathcal{T}_{\text{In}} f_2(s,a), \ \ \forall (s,a) \text{ s.t. } \hat{\beta}(a|s) > 0. \tag{55}$$

*Proof.* According to the definitions, for all $(s,a)$ s.t. $\hat{\beta}(a|s) > 0$,

$$\mathcal{T}_{\text{In}} f(s,a) = R(s,a) + \gamma \mathbb{E}_{s' \sim P(\cdot|s,a)} \left[ \max_{a' \sim \hat{\beta}(\cdot|s')} f(s', a') \right] \tag{56}$$

$f_1$ and $f_2$ satisfy

$$f_1(s,a) \geq f_2(s,a), \forall (s,a) \ \text{ s.t. } \hat{\beta}(a|s) > 0.$$

Therefore, for all $(s,a)$ s.t. $\hat{\beta}(a|s) > 0$,

$$\mathcal{T}_{\text{In}} f_1(s,a) - \mathcal{T}_{\text{In}} f_2(s,a)$$

$$= \gamma \mathbb{E}_{s' \sim P(\cdot|s,a)} \left[ \max_{a' \sim \hat{\beta}(\cdot|s')} f_1(s', a') - \max_{a' \sim \hat{\beta}(\cdot|s')} f_2(s', a') \right]$$

$$\geq 0$$

$\square$

Now we start the proof of Theorem 4 in the main paper.

We consider the iteration starting from arbitrary function $Q^0$: $\hat{Q}_{\text{DMG}}^k = \hat{\mathcal{T}}_{\text{DMG}} \hat{Q}_{\text{DMG}}^{k-1}$ and $Q_{\text{In}}^k = \mathcal{T}_{\text{In}} Q_{\text{In}}^{k-1}, \forall k \in \mathbb{Z}^+$. The possible value of $\hat{Q}_{\text{DMG}}^k$ is upper bounded by the following results.

**Theorem 9** (Limited over-estimation, Theorem 4). *Under Assumption 10, the learned Q function of DMG by iterating $\hat{\mathcal{T}}_{\text{DMG}}$ satisfies the following inequality*

$$Q_{\text{In}}^k(s,a) \leq \hat{Q}_{\text{DMG}}^k(s,a) \leq Q_{\text{In}}^k(s,a) + \frac{\lambda \epsilon_a K_Q \gamma}{1 - \gamma} (1 - \gamma^k), \ \forall s, a \sim \mathcal{D}, \ \forall k \in \mathbb{Z}^+. \tag{57}$$

*Proof.* Under worst-case generalization, $\hat{\mathcal{T}}_{\mathrm{DMG}}$ is only defined in the area $\hat{\beta}(a|s) > 0$, i.e., the dataset, and may have any generalization error at other $(s, a)$.

For any function $f$ and any $s, a \sim \mathcal{D}$,

$$
\begin{aligned}
&\hat{\mathcal{T}}_{\mathrm{DMG}} f(s, a) - \mathcal{T}_{\mathrm{In}} f(s, a) \\
=& R(s, a) + \gamma \mathbb{E}_{s' \sim P(\cdot|s,a)} \left[ \lambda \max_{a' \sim \tilde{\beta}(\cdot|s')} f(s', a') + (1 - \lambda) \max_{a' \sim \hat{\beta}(\cdot|s')} f(s', a') \right] \\
&- R(s, a) - \gamma \mathbb{E}_{s' \sim P(\cdot|s,a)} \left[ \max_{a' \sim \hat{\beta}(\cdot|s')} f(s', a') \right] \\
=& \gamma \mathbb{E}_{s' \sim P(\cdot|s,a)} \left[ \lambda \max_{a' \sim \tilde{\beta}(\cdot|s')} f(s', a') - \lambda \max_{a' \sim \hat{\beta}(\cdot|s')} f(s', a') \right] \\
\leq& \gamma \lambda \epsilon_a K_Q
\end{aligned}
$$

where the last inequality holds by Lemma 7.

On the other hand, because $\tilde{\beta}$ has a wider support than $\hat{\beta}$, we also have

$$
\hat{\mathcal{T}}_{\mathrm{DMG}} f(s, a) - \mathcal{T}_{\mathrm{In}} f(s, a) \geq 0
$$

Therefore, for any function $f$, the following inequality holds:

$$
\mathcal{T}_{\mathrm{In}} f(s, a) \leq \hat{\mathcal{T}}_{\mathrm{DMG}} f(s, a) \leq \mathcal{T}_{\mathrm{In}} f(s, a) + \gamma \lambda \epsilon_a K_Q, \quad \forall s, a \sim \mathcal{D}. \tag{58}
$$

Let $f$ in Equation (58) be $Q^0$. We have

$$
Q_{\mathrm{In}}^1(s, a) \leq \hat{Q}_{\mathrm{DMG}}^1(s, a) \leq Q_{\mathrm{In}}^1(s, a) + \frac{\lambda \epsilon_a K_Q \gamma}{1 - \gamma} (1 - \gamma), \quad \forall s, a \sim \mathcal{D}. \tag{59}
$$

This is the same as Equation (57) with $k = 1$. Therefore, Equation (57) holds when $k = 1$.

Suppose when $k = i$, Equation (57) holds:

$$
Q_{\mathrm{In}}^i(s, a) \leq \hat{Q}_{\mathrm{DMG}}^i(s, a) \leq Q_{\mathrm{In}}^i(s, a) + \frac{\lambda \epsilon_a K_Q \gamma}{1 - \gamma} (1 - \gamma^i), \quad \forall s, a \sim \mathcal{D}. \tag{60}
$$

Then let $f$ in Equation (58) be $\hat{Q}_{\mathrm{DMG}}^i$. We have

$$
\mathcal{T}_{\mathrm{In}} \hat{Q}_{\mathrm{DMG}}^i(s, a) \leq \hat{Q}_{\mathrm{DMG}}^{i+1}(s, a) = \hat{\mathcal{T}}_{\mathrm{DMG}} \hat{Q}_{\mathrm{DMG}}^i(s, a) \leq \mathcal{T}_{\mathrm{In}} \hat{Q}_{\mathrm{DMG}}^i(s, a) + \gamma \lambda \epsilon_a K_Q, \quad \forall s, a \sim \mathcal{D}. \tag{61}
$$

On the one hand, according to Lemma 8 and Equation (60), for any $s, a \sim \mathcal{D}$, we have

$$
\begin{aligned}
&\mathcal{T}_{\mathrm{In}} \hat{Q}_{\mathrm{DMG}}^i(s, a) \\
\leq& \mathcal{T}_{\mathrm{In}} \left( Q_{\mathrm{In}}^i(s, a) + \frac{\lambda \epsilon_a K_Q \gamma}{1 - \gamma} (1 - \gamma^i) \right) \\
=& R(s, a) + \gamma \mathbb{E}_{s' \sim P(\cdot|s,a)} \left[ \max_{a' \sim \hat{\beta}(\cdot|s')} \left( Q_{\mathrm{In}}^i(s', a') + \frac{\lambda \epsilon_a K_Q \gamma}{1 - \gamma} (1 - \gamma^i) \right) \right] \\
=& R(s, a) + \gamma \mathbb{E}_{s' \sim P(\cdot|s,a)} \left[ \max_{a' \sim \hat{\beta}(\cdot|s')} Q_{\mathrm{In}}^i(s', a') \right] + \gamma \frac{\lambda \epsilon_a K_Q \gamma}{1 - \gamma} (1 - \gamma^i) \\
=& \mathcal{T}_{\mathrm{In}} Q_{\mathrm{In}}^i(s, a) + \gamma \frac{\lambda \epsilon_a K_Q \gamma}{1 - \gamma} (1 - \gamma^i) \\
=& Q_{\mathrm{In}}^{i+1}(s, a) + \gamma \frac{\lambda \epsilon_a K_Q \gamma}{1 - \gamma} (1 - \gamma^i) \tag{62}
\end{aligned}
$$

Combining Equations (61) and (62), for any $s, a \sim \mathcal{D}$, we have

$$
\begin{aligned}
&\hat{Q}_{\text{DMG}}^{i+1}(s, a) \\
\leq &Q_{\text{In}}^{i+1}(s, a) + \gamma \frac{\lambda \epsilon_a K_Q \gamma}{1 - \gamma}(1 - \gamma^i) + \gamma \lambda \epsilon_a K_Q \\
= &Q_{\text{In}}^{i+1}(s, a) + \lambda \epsilon_a K_Q \gamma \left( \frac{\gamma(1 - \gamma^i)}{1 - \gamma} + 1 \right) \\
= &Q_{\text{In}}^{i+1}(s, a) + \frac{\lambda \epsilon_a K_Q \gamma}{1 - \gamma}(1 - \gamma^{i+1})
\end{aligned}
$$

On the other hand, according to Lemma 8 and Equation (60), for any $s, a \sim \mathcal{D}$, we have

$$
\mathcal{T}_{\text{In}} \hat{Q}_{\text{DMG}}^i(s, a) \geq \mathcal{T}_{\text{In}} Q_{\text{In}}^i(s, a) = Q_{\text{In}}^{i+1}(s, a) \tag{63}
$$

Combining Equations (61) and (63), for any $s, a \sim \mathcal{D}$, we have

$$
\hat{Q}_{\text{DMG}}^{i+1}(s, a) \geq Q_{\text{In}}^{i+1}(s, a). \tag{64}
$$

Hence, Equation (57) still holds when $k = i + 1$:

$$
Q_{\text{In}}^{i+1}(s, a) \leq \hat{Q}_{\text{DMG}}^{i+1}(s, a) \leq Q_{\text{In}}^{i+1}(s, a) + \frac{\lambda \epsilon_a K_Q \gamma}{1 - \gamma}(1 - \gamma^{i+1}), \ \forall s, a \sim \mathcal{D}. \tag{65}
$$

Therefore, Equation (57) holds for all $k \in \mathbb{Z}^+$, which concludes the proof. $\square$

Since in-sample training eliminates extrapolation error completely [37, 92], $Q_{\text{In}}^k$ can be considered a relatively accurate estimate. Therefore, Theorem 9 indicates that DMG has limited over-estimation under the worst generalization case. Moreover, the bound gets tighter as $\epsilon_a$ gets smaller (more mild action generalization) and $\lambda$ gets smaller (more mild generalization propagation). This is consistent with our intuitions in Section 3.2.

Finally, Theorem 5 in the main paper shows that even under worst-case generalization, DMG is guaranteed to output a safe policy with a performance lower bound.

We give a lemma before we start the proof of Theorem 5,

**Lemma 9.** *Let $\pi_1$ and $\pi_2$ be two deterministic policies. Under Assumption 11, the following inequality holds:*

$$
\text{TV}\left(d^{\pi_1} || d^{\pi_2}\right) \leq C K_P \max_s \|\pi_1(s) - \pi_2(s)\| \tag{66}
$$

*where $C$ is a positive constant and $d^\pi(s)$ is the state occupancy induced by $\pi$.*

$$
d^\pi(s) = (1 - \gamma) \sum_{t=0}^\infty \gamma^t \mathbb{E}_\pi \left[ \mathbb{I}\left[ s_t = s \right] \right]. \tag{67}
$$

*Proof.* Please refer to Lemma A.5 in [61] and Lemma 1 in [87]. $\square$

**Theorem 10** (Performance lower bound, Theorem 5). *Let $\hat{\pi}_{\text{DMG}}$ be the learned policy of DMG by iterating $\hat{\mathcal{T}}_{\text{DMG}}$, $\pi^*$ be the optimal policy, and $\epsilon_\mathcal{D}$ be the inherent performance gap of the in-sample optimal policy $\epsilon_\mathcal{D} := J(\pi^*) - J(\pi_{\text{In}}^*)$. Under Assumptions 10 and 11, for sufficiently small $\epsilon_a$, we have*

$$
J(\hat{\pi}_{\text{DMG}}) \geq J(\pi^*) - \frac{C K_P R_{\max}}{1 - \gamma} \epsilon_a - \epsilon_\mathcal{D}. \tag{68}
$$

*where $C$ is a positive constant.*

*Proof.* Following previous works [38, 83, 37, 49], we define the in-sample optimal policy as $\pi_{\text{In}}^*$:

$$
\pi_{\text{In}}^*(s) = \underset{a \sim \hat{\beta}(\cdot | s)}{\arg\max} Q_{\text{In}}^*(s, a) \tag{69}
$$

We also use $\epsilon_{\mathcal{D}}$ to denote the performance gap between the in-sample optimal policy and the globally optimal policy, which is fixed once the dataset is provided.

$$\epsilon_{\mathcal{D}} = J(\pi^*) - J(\pi^*_{\text{In}}). \tag{70}$$

We use $\hat{Q}_{\text{DMG}}$ to denote the learned Q function of DMG with sufficient iteration steps $\hat{Q}^k_{\text{DMG}}$, $k \to \infty$. And $\hat{\pi}_{\text{DMG}}$ is the output policy of $\hat{Q}_{\text{DMG}}$:

$$\hat{\pi}_{\text{DMG}}(s) = \underset{a \sim \tilde{\beta}(\cdot|s)}{\arg\max} \, \hat{Q}_{\text{DMG}}(s, a) \tag{71}$$

It holds that

$$
\begin{aligned}
&|J(\pi^*) - J(\hat{\pi}_{\text{DMG}})| \\
=&|J(\pi^*) - J(\pi^*_{\text{In}}) + J(\pi^*_{\text{In}}) - J(\hat{\pi}_{\text{DMG}})| \\
\leq&|J(\pi^*) - J(\pi^*_{\text{In}})| + |J(\pi^*_{\text{In}}) - J(\hat{\pi}_{\text{DMG}})| \\
=&\epsilon_{\mathcal{D}} + |J(\pi^*_{\text{In}}) - J(\hat{\pi}_{\text{DMG}})|
\end{aligned} \tag{72}
$$

In the following, we bound the term $|J(\pi^*_{\text{In}}) - J(\hat{\pi}_{\text{DMG}})|$.

$$
\begin{aligned}
&|J(\pi^*_{\text{In}}) - J(\hat{\pi}_{\text{DMG}})| \\
=&\left| \frac{1}{1-\gamma} \mathbb{E}_{s \sim d^{\hat{\pi}_{\text{DMG}}}}[r(s)] - \frac{1}{1-\gamma} \mathbb{E}_{s \sim d^{\pi^*_{\text{In}}}}[r(s)] \right| \\
=&\frac{1}{1-\gamma} \left| \sum_s \left( d^{\hat{\pi}_{\text{DMG}}}(s) - d^{\pi^*_{\text{In}}}(s) \right) r(s) \right| \\
\leq&\frac{1}{1-\gamma} \sum_s \left| \left( d^{\hat{\pi}_{\text{DMG}}}(s) - d^{\pi^*_{\text{In}}}(s) \right) \right| |r(s)| \\
\leq&\frac{R_{\max}}{1-\gamma} \text{TV} \left( d^{\hat{\pi}_{\text{DMG}}}(s) || d^{\pi^*_{\text{In}}}(s) \right) \\
\leq&\frac{R_{\max}}{1-\gamma} C K_P \max_s \| \hat{\pi}_{\text{DMG}}(s) - \pi^*_{\text{In}}(s) \|
\end{aligned} \tag{73}
$$

where the last inequality holds by Lemma 9.

According to Theorem 9, $\hat{Q}_{\text{DMG}}$ satisfies the following inequality:

$$Q^*_{\text{In}}(s, a) \leq \hat{Q}_{\text{DMG}}(s, a) \leq Q^*_{\text{In}}(s, a) + \frac{\lambda \epsilon_a K_Q \gamma}{1 - \gamma}, \ \forall s, a \sim \mathcal{D}. \tag{74}$$

It means that for any $(s, a) \sim \mathcal{D}$, with sufficiently small $\epsilon_a$, $\hat{Q}_{\text{DMG}}(s, a)$ sufficiently approximates $Q^*_{\text{In}}(s, a)$. By Definition 4, $\tilde{\beta}$ is a mildly generalized policy. That is, for any $s \sim \mathcal{D}$, $\tilde{\beta}$ satisfies

$$\text{supp}(\hat{\beta}(\cdot|s)) \subseteq \text{supp}(\tilde{\beta}(\cdot|s)), \ \text{and} \ \max_{a_1 \sim \tilde{\beta}(\cdot|s)} \min_{a_2 \sim \hat{\beta}(\cdot|s)} \|a_1 - a_2\| \leq \epsilon_a,$$

As $\hat{\pi}_{\text{DMG}}(s) \in \tilde{\beta}(\cdot|s)$, it implies that we can find $a_{\text{in}} \in \hat{\beta}(\cdot|s)$ (in dataset) such that $\| \hat{\pi}_{\text{DMG}}(s) - a_{\text{in}} \| \leq \epsilon_a$.

Now suppose $a_{\text{in}}$ is not the maximum point of $Q^*_{\text{In}}(s, \cdot)$ at a certain $s$. We use $\pi^*_{\text{In}}(s)$ to denote the maximum point of $Q^*_{\text{In}}(s, \cdot)$. Let $\epsilon_{Q^*_{\text{In}}}$ be the gap between $Q^*_{\text{In}}(s, a_{\text{in}})$ and $Q^*_{\text{In}}(s, \pi^*_{\text{In}}(s))$:

$$\epsilon_{Q^*_{\text{In}}}(s) := Q^*_{\text{In}}(s, \pi^*_{\text{In}}(s)) - Q^*_{\text{In}}(s, a_{\text{in}}) > 0. \tag{75}$$

By Assumption 10 (Lipschitz $Q$), we have

$$\hat{Q}_{\text{DMG}}(s, \hat{\pi}_{\text{DMG}}(s)) - \hat{Q}_{\text{DMG}}(s, a_{\text{in}}) \leq K_Q \| \hat{\pi}_{\text{DMG}}(s) - a_{\text{in}} \| \leq K_Q \epsilon_a. \tag{76}$$

Therefore,

$$\hat{Q}_{\mathrm{DMG}}(s, \pi_{\mathrm{In}}^*(s)) - \hat{Q}_{\mathrm{DMG}}(s, \hat{\pi}_{\mathrm{DMG}}(s))$$
$$\geq \hat{Q}_{\mathrm{DMG}}(s, \pi_{\mathrm{In}}^*(s)) - \hat{Q}_{\mathrm{DMG}}(s, a_{\mathrm{in}}) - K_Q \epsilon_a$$
$$\geq Q_{\mathrm{In}}^*(s, \pi_{\mathrm{In}}^*(s)) - Q_{\mathrm{In}}^*(s, a_{\mathrm{in}}) - \frac{\lambda \epsilon_a K_Q \gamma}{1 - \gamma} - K_Q \epsilon_a$$
$$= \epsilon_{Q_{\mathrm{In}}^*}(s) - \frac{\lambda \epsilon_a K_Q \gamma}{1 - \gamma} - K_Q \epsilon_a$$

where the first inequality holds by Equation (76), the second inequality holds by Equation (74), and the last equality holds by Equation (75).

Hence, for sufficiently small $\epsilon_a$ such that $\epsilon_{Q_{\mathrm{In}}^*}(s) - \frac{\lambda \epsilon_a K_Q \gamma}{1-\gamma} - K_Q \epsilon_a > 0$, i.e.,

$$\epsilon_a < \frac{(1-\gamma)\epsilon_{Q_{\mathrm{In}}^*}(s)}{K_Q(1 - \gamma + \lambda\gamma)}, \tag{77}$$

it holds that $\hat{Q}_{\mathrm{DMG}}(s, \pi_{\mathrm{In}}^*(s)) - \hat{Q}_{\mathrm{DMG}}(s, \hat{\pi}_{\mathrm{DMG}}(s)) > 0$. As $\pi_{\mathrm{In}}^*(s) \in \hat{\beta}(\cdot|s)$, it also satisfies $\pi_{\mathrm{In}}^*(s) \in \tilde{\beta}(\cdot|s)$. This contradicts the definition of $\hat{\pi}_{\mathrm{DMG}}(s)$ in Equation (71):

$$\hat{\pi}_{\mathrm{DMG}}(s) = \underset{a \sim \tilde{\beta}(\cdot|s)}{\operatorname{argmax}} \hat{Q}_{\mathrm{DMG}}(s, a)$$

Therefore, $a_{\mathrm{in}}$ is the maximum point of $Q_{\mathrm{In}}^*(s, \cdot)$. In other words, the maximum point of $Q_{\mathrm{In}}^*(s, \cdot)$ (denoted by $\pi_{\mathrm{In}}^*(s)$) is the closest neighbor of $\hat{\pi}_{\mathrm{DMG}}(s)$ in the dataset ($\hat{\beta}(\cdot|s) > 0$):

$$\pi_{\mathrm{In}}^*(s) = \underset{a \sim \hat{\beta}(\cdot|s)}{\operatorname{argmin}} \|a - \hat{\pi}_{\mathrm{DMG}}(s)\|$$

As $\hat{\pi}_{\mathrm{DMG}}(s) \in \tilde{\beta}(\cdot|s)$, the following inequality holds by Definition 4:

$$\|\hat{\pi}_{\mathrm{DMG}}(s) - \pi_{\mathrm{In}}^*(s)\| \leq \epsilon_a.$$

Therefore, we have

$$|J(\pi_{\mathrm{In}}^*) - J(\hat{\pi}_{\mathrm{DMG}})| \leq \frac{R_{\max}}{1 - \gamma} C K_P \epsilon_a. \tag{78}$$

By combining Equations (72) and (78), we have

$$J(\hat{\pi}_{\mathrm{DMG}}) \geq J(\pi^*) - \frac{C K_P R_{\max}}{1 - \gamma} \epsilon_a - \epsilon_{\mathcal{D}}. \tag{79}$$

This concludes the proof. $\qquad\square$

## C  Experimental Details

### C.1  Experimental Details in Offline Experiments

Our evaluation criteria follow those used in most previous works. For the Gym locomotion tasks, we average returns over 10 evaluation trajectories and 5 random seeds, while for the AntMaze tasks, we average over 100 evaluation trajectories and 5 random seeds. Following the suggestions in the benchmark [16], we subtract 1 from the rewards for the AntMaze datasets. And following previous works [17, 37, 83, 88], we normalize the states in Gym locomotion datasets. We choose TD3 [18] as our base algorithm and optimize a deterministic policy. Thus we replace the log likelihood in Eq. (14) with mean squared error in practice, which is equivalent to optimizing a Gaussian policy with fixed variance [17]. The reported results are the normalized scores, which are offered by the D4RL benchmark [16] to measure how the learned policy compared with random and expert policy:

$$\text{D4RL score} = 100 \times \frac{\text{learned policy return} - \text{random policy return}}{\text{expert policy return} - \text{random policy return}}$$

Table 5: Hyperparameters of DMG.

| | Hyperparameter | Value |
|---|---|---|
| DMG | Optimizer | Adam [34] |
| | Critic learning rate | $3 \times 10^{-4}$ |
| | Actor learning rate | $3 \times 10^{-4}$ with cosine schedule |
| | Batch size | 256 |
| | Discount factor | 0.99 |
| | Number of iterations | $10^6$ |
| | Target update rate | 0.005 |
| | Number of Critics | 2 |
| | Penalty coefficient $\nu$ | {0.1,10} for Gym-MuJoCo
{0.5} for Antmaze |
| | Mixture coefficient $\lambda$ | 0.25 |
| IQL Specific | Expectile $\tau$ | 0.7 for Gym-MuJoCo
0.9 for Antmaze |
| | Inverse temperature $\alpha$ | 3.0 for Gym-MuJoCo
10.0 for Antmaze |
| Architecture | Actor | input-256-256-output |
| | Critic | input-256-256-1 |

As we implement our main algorithm based on IQL [37], we use the hyperparameters suggested in their paper for fair comparisons, i.e., $\tau = 0.7$ and $\alpha = 3$ for Gym locomotion tasks and $\tau = 0.9$ and $\alpha = 10$ for AntMaze tasks. For the results of $\mathcal{X}$QL+DMG and SQL+DMG, we also adopt the suggested hyperparameters in their papers [21, 88] for fair comparisons. In detail, we choose $\beta$ in $\mathcal{X}$QL [21] as $5.0$ in medium, medium-replay, and medium-expert datasets, and $\alpha$ in SQL [88] as $2.0$ for medium, medium-replay datasets, and $5.0$ for medium-expert datasets.

DMG has two main hyperparameters: mixture coefficient $\lambda$ and penalty coefficient $\nu$. We use $\lambda = 0.25$ for all tasks. We use $\nu = 0.5$ for Antmaze tasks and $\nu \in \{0.1, 10\}$ for Gym locomotion tasks (0.1 for medium, medium-replay, random datasets; 10 for expert and medium-expert datasets). All hyperparameters of DMG are included in Table 5.

## C.2 Experimental Details in Offline-to-online Experiments

For online fine-tuning experiments, we first run offline RL for $1 \times 10^6$ gradient steps. Then we continue training while collecting data actively in the environment and adding the data to the replay buffer. We perform online fine-tuning for $1 \times 10^6$ steps with 1 update-to-data (UTD) ratio, and collect data with exploration noise 0.1 as suggested by TD3 [18]. During offline pre-training, we fix the mixture coefficient $\lambda = 0.25$ and the penalty coefficient $\nu = 0.5$, while in the online phase, we exponentially adjust $\lambda$ and $\nu$, as DMG with $\lambda = 1$ and $\nu = 0$ corresponds to standard online RL. In the challenging AntMaze domains characterized by high-dimensional state and action spaces, as well as sparse rewards, the extrapolation error remains significant even during the online phase [83]. Therefore, we decay $\lambda$ from 0.25 to 0.5 and $\nu$ from 0.5 to 0.005 (1% of its initial value), employing a decay rate of 0.99 every 1000 gradient steps. Additionally, following previous works [83, 72], we set $\gamma = 0.995$ when fine-tuning on antmaze-large datasets, for both DMG and IQL to ensure a fair comparison. All other training details remain consistent between the offline RL phase and the online fine-tuning phase.

# D Additional Experimental Results

## D.1 Computational Cost

We test the runtime of offline RL algorithms on halfcheetah-medium-replay-v2 on a GeForce RTX 3090. The results of DMG and other baselines are shown in Figure 3. It takes 1.7h for DMG to finish the task, which is comparable to the fastest offline RL algorithm TD3BC [17].

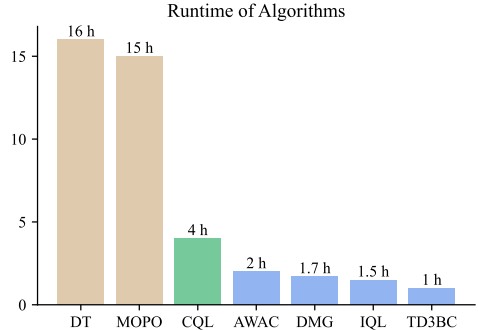

Figure 3: Runtime of algorithms on halfcheetah-medium-replay-v2 on a GeForce RTX 3090.

## D.2 Offline Training Results of DMG on More Random Seeds

The experimental results in the main paper show the mean and standard deviation (SD) over five random seeds. According to [56], we conduct experiments to test DMG on additional random seeds, reporting 95% confidence interval (CI) over 10 random seeds. Table 6 shows the comparison between the new results (10seeds/95%CI) and the previously reported results (5seeds/SD in Table 2) on the D4RL offline training tasks. The results show that our method achieves about the same performance as under the previous evaluation criterion.

Table 6: Comparison of DMG under different evaluation criteria on D4RL offline training tasks.

| Dataset-v2 | DMG (5seeds/SD) | DMG (10seeds/95%CI) |
|---|---|---|
| halfcheetah-m | **54.9±0.2** | **54.9±0.3** |
| hopper-m | **100.6±1.9** | 100.5±1.0 |
| walker2d-m | **92.4±2.7** | 92.0±1.2 |
| halfcheetah-m-r | **51.4±0.3** | **51.4±0.4** |
| hopper-m-r | 101.9±1.4 | **102.1±0.6** |
| walker2d-m-r | 89.7±5.0 | **90.3±2.8** |
| halfcheetah-m-e | 91.1±4.2 | **92.9±2.1** |
| hopper-m-e | **110.4±3.4** | 109.0±2.6 |
| walker2d-m-e | **114.4±0.7** | 113.9±1.2 |
| halfcheetah-e | **95.9±0.3** | **95.9±0.2** |
| hopper-e | 111.5±2.2 | **111.8±1.3** |
| walker2d-e | **114.7±0.4** | 114.5±0.3 |
| halfcheetah-r | **28.8±1.3** | 28.7±1.2 |
| hopper-r | 20.4±10.4 | **21.6±6.6** |
| walker2d-r | 4.8±2.2 | **7.7±3.0** |
| locomotion total | 1182.8 | **1187.2** |
| antmaze-u | **92.4±1.8** | 91.8±1.6 |
| antmaze-u-d | **75.4±8.1** | 73.0±5.0 |
| antmaze-m-p | 80.2±5.1 | **80.5±2.1** |
| antmaze-m-d | **77.2±6.1** | 76.7±3.6 |
| antmaze-l-p | 55.4±6.2 | **56.7±3.6** |
| antmaze-l-d | **58.8±4.5** | 57.2±2.7 |
| antmaze total | **439.4** | 435.9 |

## D.3 Learning Curves of DMG during Offline Training

Learning curves during offline training on Gym-MuJoCo locomotion tasks and Antmaze tasks are presented in Figure 4 and Figure 5, respectively. The curves are averaged over 5 random seeds, with the shaded area representing the standard deviation across seeds.

## D.4   Learning Curves of DMG during Online Fine-tuning

Learning curves during online fine-tuning on Antmaze tasks are presented in Figure 6. The curves are averaged over 5 random seeds, with the shaded area representing the standard deviation across seeds.

# E   Broader Impact

Offline reinforcement learning (RL) presents a promising avenue for enhancing and broadening the practical applicability of RL across various domains including robotics, recommendation systems, healthcare, and education, characterized by costly or hazardous data collection processes. However, it is imperative to recognize the potential adverse societal ramifications associated with any offline RL algorithm. One such concern pertains to the possibility that the offline data utilized for training may harbor inherent biases, which could subsequently permeate into the acquired policy. Furthermore, it is essential to contemplate the potential implications of offline RL on employment, given its contribution to automating tasks conventionally executed by human experts, such as factory automation or autonomous driving. Addressing these challenges is essential for fostering the responsible development and deployment of offline RL algorithms, with the aim of maximizing their positive impact while mitigating negative societal consequences.

From an academic perspective, this research scrutinizes offline RL through the lens of generalization, balancing the need for generalization with the risk of over-generalization. The proposed approach DMG potentially offers researchers a new perspective on appropriately exploiting generalization in offline RL. Besides, DMG also holds the promise to be extended to safe RL [1, 26, 20], multi-agent RL [43, 62, 65, 60, 25], and meta RL [15, 76, 77, 75, 4].

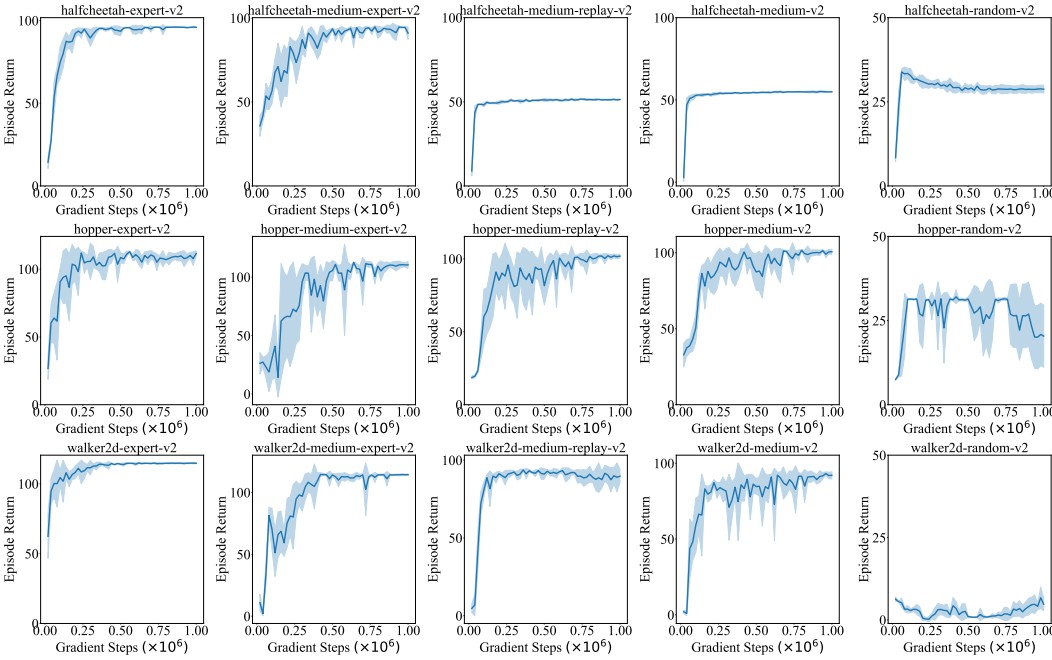

Figure 4: Learning curves of DMG on Gym locomotion tasks during offline training. The curves are averaged over 5 random seeds, with the shaded area representing the standard deviation across seeds.

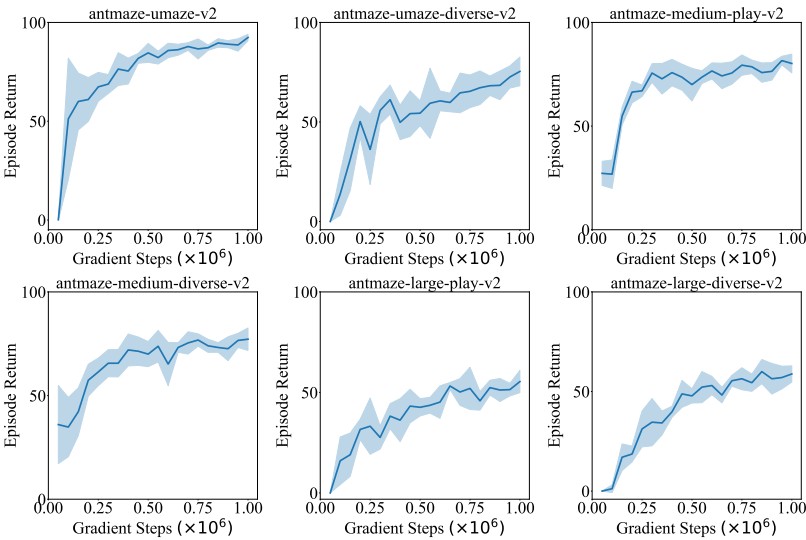

Figure 5: Learning curves of DMG on Antmaze tasks during offline training. The curves are averaged over 5 random seeds, with the shaded area representing the standard deviation across seeds.

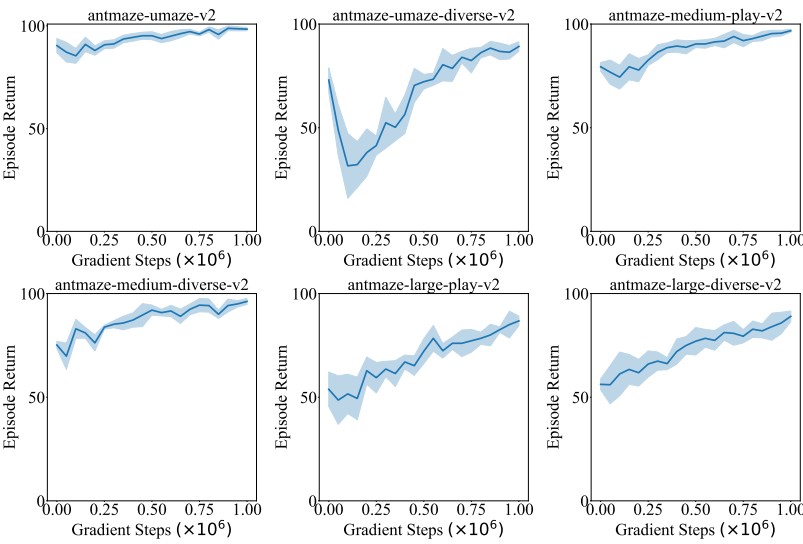

Figure 6: Learning curves of DMG on Antmaze tasks during online fine-tuning. The curves are averaged over 5 random seeds, with the shaded area representing the standard deviation across seeds.

