# OpenReview forum: "Doubly Mild Generalization for Offline Reinforcement Learning"
_NeurIPS.cc/2024/Conference — NeurIPS 2024 poster_

### Official Review · Reviewer_iL9e · 2024-07-02

**Soundness:** 3
**Presentation:** 4
**Contribution:** 3
**Rating:** 6
**Confidence:** 4

**Summary:**

The paper proposes an algorithm to allow for some mild generalization to OOD actions in offline RL by proposing constraints on which actions to generalize on and by constraining bootstrapping signal to avoid overestimation. It includes results and ablations on common offline RL benchmarks.

**Strengths:**

1. The idea is very simple (which is a good thing!), intuitive, and can be easily integrated into existing algorithms. I think it provides nice insight into how to think about generalization and overestimation which can be useful for others.
2. The paper does a nice job of explaining the intuition behind the method and faults with previous methods. I appreciated Table 1.
3. The paper includes results on downstream performance metrics as well as ablation studies.

**Weaknesses:**

1. It is a bit concerning that the main results are over only 5 seeds. While it is commonly done in prior work, I don’t think that means it's correct. I’d refer authors to [1]. I think it would be nice to increase the number of seeds.
2. Related to above, the variations seem to be based on standard errors (for example, Table 2) but it has typically been documented that these are unreliable and should be avoided [1] in RL research. Instead opting for some high-confidence confidence interval would be better.

I would be willing to increase the score if the results were still true with more seeds.

[1] Empirical Design in Reinforcement Learning. Patterson et al. 2023.

**Questions:**

1. How do the authors think the insights in this paper can be used for off-policy evaluation where the behavior policy is unknown and we want to evaluate a fixed target policy? The control setting allows some flexibility in say , constraining what the optimal policy can be, but with evaluation this seems trickier.
2. The method seems to rely on using the empirical behavior policy to constrain the policy improvement. This seems reasonable, but the paper does not study this deeply. While this question may seem separate from the question investigated in this paper, in offline RL, it is typical for multiple policies to generate the data, which means the empirical policy is a multi-modal policy. I am curious how sensitive the proposed method would be to bad estimates in the empirical policy? At the very least I think the paper should discuss this point since the algorithm makes the assumption that such an empirical policy is easily accessible.

**Limitations:**

Yes, the paper addresses this in Section 6.

---

> ### Author Rebuttal · Authors · 2024-08-07
>
> We appreciate the time and effort you are dedicated to providing feedback on our paper and are grateful for the meaningful comments.
>
> **Q1: More random seeds and use of confidence interval in evaluation.**
>
> Thanks for the kind suggestions and this nice reference. [1] provides a comprehensive resource for how to do good experiments in RL. We have taken your advice and conducted experiments on 10 random seeds, reporting results based on a 95% confidence interval (CI). The following table shows the comparison between the newly obtained results (10seeds/95%CI) and the previously reported results (5seeds/SD). We also present the new learning curves on all the tasks in **Figures 1 and 2** of the PDF (attached to the global response). The results show that our method achieves about the same performance as under the previous evaluation criterion. In the latter revision, we will cite [1] and update the results accordingly.
>
> Table 1: Comparison of DMG under different evaluation criteria.
>
> | Dataset | DMG (5seeds/SD) | DMG (10seeds/95%CI) |
> | --- | --- | --- |
> | halfcheetah-m | **54.9$\pm$0.2** | **54.9$\pm$0.3** |
> | hopper-m | **100.6$\pm$1.9** | 100.5$\pm$1.0 |
> | walker2d-m | **92.4$\pm$2.7** | 92.0$\pm$1.2 |
> | halfcheetah-m-r | **51.4$\pm$0.3** | **51.4$\pm$0.4** |
> | hopper-m-r | 101.9$\pm$1.4 | **102.1$\pm$0.6** |
> | walker2d-m-r | 89.7$\pm$5.0 | **90.3$\pm$2.8** |
> | halfcheetah-m-e | 91.1$\pm$4.2 | **92.9$\pm$2.1** |
> | hopper-m-e | **110.4$\pm$3.4** | 109.0$\pm$2.6 |
> | walker2d-m-e | **114.4$\pm$0.7** | 113.9$\pm$1.2 |
> | halfcheetah-e | **95.9$\pm$0.3** | **95.9$\pm$0.2** |
> | hopper-e | 111.5$\pm$2.2 | **111.8$\pm$1.3** |
> | walker2d-e | **114.7$\pm$0.4** | 114.5$\pm$0.3 |
> | halfcheetah-r | **28.8$\pm$1.3** | 28.7$\pm$1.2 |
> | hopper-r | 20.4$\pm$10.4 | **21.6$\pm$6.6** |
> | walker2d-r | 4.8$\pm$2.2 | **7.7$\pm$3.0** |
> | locomotion total | 1182.8 | **1187.2** |
> |  |  |  |
> | antmaze-u | **92.4$\pm$1.8** | 91.8$\pm$1.6 |
> | antmaze-u-d | **75.4$\pm$8.1** | 73.0$\pm$5.0 |
> | antmaze-m-p | 80.2$\pm$5.1 | **80.5$\pm$2.1** |
> | antmaze-m-d | **77.2$\pm$6.1** | 76.7$\pm$3.6 |
> | antmaze-l-p | 55.4$\pm$6.2 | **56.7$\pm$3.6** |
> | antmaze-l-d | **58.8$\pm$4.5** | 57.2$\pm$2.7 |
> | antmaze total | **439.4** | 435.9 |
>
> **Q2: How do the authors think the insights in this paper can be used for off-policy evaluation where the behavior policy is unknown and we want to evaluate a fixed target policy?**
>
> Thanks for this insightful question. Indeed, the evaluation setting appears trickier because the target policy is fixed. While mild action generalization no longer seems applicable in this scenario, mild generalization propagation still applies. Its direct analogue in the off-policy evaluation scenario is that: the Bellman target uses a mixture of (1) the Q-value of the action outputted by the target policy and (2) the Q-value of the nearest neighbor of that action in the dataset (or the replay buffer). This approach brings bias on the one hand, but can also be expected to have a smaller variance. An in-depth analysis of this would be an interesting direction for future work.
>
> **Q3: The method seems to rely on using the empirical behavior policy to constrain the policy improvement.**
>
> We apologize for the confusion. Actually, our method does not require training an empirical behavior policy. The implementation of policy improvement is discussed in Section 3.5 of the paper. Here, we first define a reshaped empirical behavior policy $\hat\beta^*$. Then we enforce the proximity between the trained policy and the reshaped behavior policy in Eq. (13). A special design worth noting is that we use the forward KL divergence $\mathrm{KL}(\hat\beta^*(\cdot|s) \| \pi_\phi(\cdot|s))$, which not only allows $\pi$ to select actions outside the support of $\hat\beta^*$ but also eliminate the need for pretraining an empirical behavior policy. Specifically, by substituting the analytical expression of $\hat\beta^*$ into the KL divergence and using importance sampling (the same derivation as in AWR [2]), Eq. (14) eliminates the need for pre-training a behavior model (maybe multi-modal as you mentioned) and enables direct sampling from the dataset for optimization.
>
> **Reference**
>
> [1] Patterson, Andrew, et al. "Empirical design in reinforcement learning." arXiv preprint arXiv:2304.01315 (2023).
>
> [2] Peng, Xue Bin, et al. "Advantage-weighted regression: Simple and scalable off-policy reinforcement learning." arXiv preprint arXiv:1910.00177 (2019).

---

> > ### Comment · Reviewer_iL9e · 2024-08-08
> >
> > Thank you for being clear in your response and addressing my concerns. Please do update the paper with the new seed and CI results. I have updated my score.

---

> > > ### Author Response · Authors · 2024-08-08
> > >
> > > Thank you for the valuable suggestion and we will make sure to update the paper with the 10seed/95%CI results. We sincerely appreciate your time and effort in reviewing our work.

---

### Official Review · Reviewer_Kscy · 2024-07-12

**Soundness:** 3
**Presentation:** 4
**Contribution:** 3
**Rating:** 7
**Confidence:** 4

**Summary:**

The paper introduces a novel approach called Doubly Mild Generalization (DMG) to tackle the well-known issue of over-generalization in Offline Reinforcement Learning (RL). Offline RL often suffers from extrapolation errors and value overestimation due to the generalization of value functions or policies towards out-of-distribution (OOD) actions. To address these challenges, the authors propose DMG, which comprises two key components: (i) mild action generalization and (ii) mild generalization propagation. Mild action generalization involves selecting actions in the close neighborhood of the dataset to maximize the Q values, while mild generalization propagation mitigates the propagation of potential erroneous generalization through bootstrapping without impeding the propagation of RL learning signals. Theoretically, DMG ensures better performance than in-sample optimal policies under oracle generalization scenarios and controls value overestimation even in worst-case generalization scenarios. Empirically, DMG achieves state-of-the-art performance on standard offline RL benchmarks, including Gym-MuJoCo locomotion tasks and challenging AntMaze tasks, and demonstrates strong online fine-tuning performance.

**Strengths:**

- **Innovative Approach**: The concept of doubly mild generalization is novel and addresses a significant challenge in offline RL by balancing the need for generalization with the risk of over-generalization.
- **Theoretical Rigor**: The paper provides a thorough theoretical analysis of DMG, proving its advantages in both oracle and worst-case generalization scenarios. The theoretical guarantees add robustness to the proposed method.
- **Empirical Performance**: DMG demonstrates superior empirical results, outperforming existing methods on multiple benchmarks. This includes both standard locomotion tasks and more complex navigation tasks, showcasing the versatility and effectiveness of the approach.
- **Seamless Transition to Online Learning**: The method's ability to smoothly transition from offline to online learning, achieving strong fine-tuning performance, is a notable strength, making it practical for real-world applications where a mixture of offline data and online interactions is common.

**Weaknesses:**

- **Dependence on Continuity Assumptions**: The theoretical analysis relies on certain continuity assumptions about the Q function and transition dynamics. While these assumptions are standard, they may not hold in all practical scenarios, potentially limiting the applicability of the theoretical guarantees.
- **Potential Sensitivity to Hyperparameters**: The performance of DMG might be sensitive to the choice of hyperparameters such as the mixture coefficient (λ) and penalty coefficient (ν). A more detailed analysis of hyperparameter sensitivity would be beneficial to understand the robustness of the method.

**Questions:**

- Can you explain why using max and min in Eq.5? It’s a bit confusing as the policy is normally a Gaussian or deterministic policy.
- What about directly regularizing the KL between  $\pi$ and $\hat \beta$ in Eq 13?
- Comparison with more SOTA algorithms such as STR is suggested.
- What does the symbol × mean in Figure 1
- It seems the performance of DMG  when $\lambda=1$ in Figure 1 is inferior to that of TD3BC. I’m curious what’s the potential reason for this phenomenon as it should be better or similar to TD3BC when the in-sample term is removed.

**Limitations:**

The authors have discussed the limitations in the draft.

---

> ### Author Rebuttal · Authors · 2024-08-07
>
> We appreciate the time and effort you are dedicated to providing feedback on our paper and are grateful for the meaningful comments.
>
> **Q1: Dependence on Continuity Assumptions.**
>
> Thanks for the comment. Indeed, our work relies on continuity assumptions about the learned Q function and transition dynamics. For the former, a continuous learned Q function can be relatively easily satisfied and is particularly necessary for the analysis of value function generalization. For the latter, continuous transition dynamics is also a standard assumption in the theoretical studies of RL [1,2,3,4,5]. Several previous works assume the transition to be Lipschitz continuous with respect to (w.r.t) both state and action [1,2]. In our paper, we need the Lipschitz continuity to hold only w.r.t. action.
>
> **Q2: Potential Sensitivity to Hyperparameters**
>
> Thanks for the comment. We will answer this question from two aspects: the robustness of the DMG principle and the robustness of the specific DMG algorithm. We have conducted an ablation study on the mixture coefficient $\lambda$ and penalty coefficient $\nu$ in Section 5.3. As shown in Figures 1 and 2, DMG generally achieves the best performance with $\lambda \in (0,1)$ and a moderate $\nu$. This demonstrates that our DMG principle generally outperforms previous non/full generalization propagation and non/full action generalization principles, which proves the robustness of the DMG principle. On the other hand, our specific DMG algorithm also involves less hyperparameter tuning overall compared to other algorithms. As stated in Appendix C1, we use $\lambda=0.25$ for all tasks, and use $\nu=0.5$ for Antmaze, $\nu \in \{0.1, 10\}$ for Gym locomotion ($0.1$ for medium, medium-replay, random datasets; $10$ for expert and medium-expert datasets)。This demonstrates the robustness of the specific DMG algorithm.
>
> **Q3: Can you explain why using max and min in Eq. (5)?**
>
> We apologize for the confusion. Eq. (5) defines mildly generalized policy, where $\tilde \beta$ and $\hat \beta$ can be any policy beyond Gaussian or deterministic policy. The max and min in Eq. (5) mean that for any $a_1 \sim \tilde \beta(\cdot|s)$ (i.e., from the defined mildly generalized policy), we can find $a_2 \sim \hat \beta(\cdot|s)$ (i.e., from the dataset) such that $\|a_1-a_2\| \leq \epsilon_a$. In other words, in a certain state, the distance between actions taken by $\tilde \beta$ and its closest action in the dataset is bounded by $\epsilon_a$.
>
> **Q4: What about directly regularizing the KL between  $\pi$ and $\hat \beta$ in Eq. (13)?**
>
> Thanks for the question. This is equivalent to setting the inverse temperature $\alpha$ as 0. The comparisons between DMG ($\alpha=0$) and the original DMG on Gym locomotion tasks are shown in the following table. Compared to DMG, DMG ($\alpha=0$) exhibits only a small drop in performance. Therefore, DMG is robust to this hyperparameter, and the choice of aligning $\pi$ and $\hat \beta^*$ in implementation is not a key factor in DMG's good performance.
>
> Table 1: Averaged normalized scores on Gym locomotion tasks over 5 random seeds.
>
> | Dataset | DMG ($\alpha=0$) | DMG |
> | --- | --- | --- |
> | halfcheetah-m | 53.6 | **54.9** |
> | hopper-m | 90.1 | **100.6** |
> | walker2d-m | 91.0 | **92.4** |
> | halfcheetah-m-r | 50.7 | **51.4** |
> | hopper-m-r | 97.4 | **101.9** |
> | walker2d-m-r | **93.3** | 89.7 |
> | halfcheetah-m-e | **93.3** | 91.1 |
> | hopper-m-e | 106.5 | **110.4** |
> | walker2d-m-e | 110.8 | **114.4** |
> | total | 786.7 | **806.8** |
>
> **Q4: Comparison with more SOTA algorithms such as STR is suggested.**
>
> Thanks for the suggestion. Due to the page limit, we report the overall performance of STR (taken from its paper) and DMG in the following table. We will include the full comparison in our latter revision. Overall, DMG achieves slightly higher performance in both Gym locomotion and Antmaze domains.
>
> Table 2: Total scores on Gym locomotion and Antmaze tasks, averaged over 5 random seeds.
>
> | Dataset | STR | DMG |
> | --- | --- | --- |
> | locomotion total | 1162.2 | **1182.8** |
> | antmaze total | 430.2 | **439.4** |
>
> **Q5: What does the symbol × mean in Figure 1?**
>
> Sorry for the confusion. The crosses (x) in Figure 1 mean that the value functions diverge in several seeds.
>
> **Q6: It seems the performance of DMG when $\lambda=1$ in Figure 1 is inferior to that of TD3BC.**
>
> When $\lambda=1$, DMG, just like TD3BC, allows full generalization propagation in value training. However, policy training and the specific hyperparameter configuration of DMG still differ from TD3BC. We hypothesize this may cause the discrepancy.
>
> **Reference**
>
> [1] Dufour, Francois, and Tomas Prieto-Rumeau. "Finite linear programming approximations of constrained discounted Markov decision processes." SIAM Journal on Control and Optimization 51.2 (2013): 1298-1324.
>
> [2] Dufour, Francois, and Tomas Prieto-Rumeau. "Approximation of average cost Markov decision processes using empirical distributions and concentration inequalities." Stochastics An International Journal of Probability and Stochastic Processes 87.2 (2015): 273-307.
>
> [3] Shah, Devavrat, and Qiaomin Xie. "Q-learning with nearest neighbors." Advances in Neural Information Processing Systems 31 (2018).
>
> [4] Xiong, Huaqing, et al. "Deterministic policy gradient: Convergence analysis." Uncertainty in Artificial Intelligence. PMLR, 2022.
>
> [5] Ran, Yuhang, et al. "Policy regularization with dataset constraint for offline reinforcement learning." International Conference on Machine Learning. PMLR, 2023.

---

> > ### Comment · Reviewer_Kscy · 2024-08-08
> >
> > I thank the authors for their response that address my concerns, I have updated my score accordingly.

---

> > > ### Author Response · Authors · 2024-08-08
> > >
> > > Thank you for your positive feedback! We sincerely appreciate the time and effort you’ve dedicated to reviewing our work.

---

### Official Review · Reviewer_RKt1 · 2024-07-12

**Soundness:** 4
**Presentation:** 3
**Contribution:** 3
**Rating:** 6
**Confidence:** 4

**Summary:**

The offline RL community has recently shown a surge of interest in in-sample offline reinforcement learning algorithms. However, these algorithms can sometimes be too restrictive, unable to leverage the generalization capability of deep neural networks. As a remedy, the authors proposed an algorithm called Doubly Mild Generalization that uses the weighted mean of the in-sample maximum and mildly generalizable maximum Q-value. Experiments conducted on various offline RL benchmarks manifest the effectiveness of the proposed algorithm.

**Strengths:**

The authors provide rigorous mathematical proofs for most of their arguments. They also provide a practical version of their algorithm, which performs very well on multiple offline RL benchmarks.

**Weaknesses:**

1. The constants $C_1$ and $C_2$ of Theorem 1 are state-dependent.

2. Lines 120-121 are difficult to understand. How does (4) relate to the generalizability of Q functions?

3. The meaning of Assumption 1 is unclear at first glance. An explanation of what might prevent $\mathcal{T}_{\textrm{DMG}}$ from being well-defined would be helpful for the readers. Also, I think the term "well-defined" is misleading. There's no problem with "defining" the DMG operator for state-action pairs outside the dataset; it's just that they would be incorrect and not be very meaningful.

### Minor comments

1. $\alpha$ is missing from Eq. (21)

2. $k=\infty$ → $k\to\infty$ in Equations (48) and (49)

**Questions:**

Please refer to the **Weaknesses** section.

**Limitations:**

The authors adequately addressed the limitations and potential negative societal impact of their work.

---

> ### Author Rebuttal · Authors · 2024-08-07
>
> We appreciate the time and effort you are dedicated to providing feedback on our paper and are grateful for the meaningful comments.
>
> **Q1: The constants $C_1$ and $C_2$ of Theorem 1 are state-dependent.**
>
> Indeed, they are state-dependent. It is worth noting that Theorem 1 provides insight and motivates our method.
>
> **Q2: How does (4) relate to the generalizability of Q functions?**
>
> We apologize for the confusion caused by the lack of sufficient explanation. Eq. (4) is the update of the parametric Q function ($Q_{\theta} \rightarrow Q_{\theta’}$) at state-action pairs $(s,\tilde a) \notin \mathcal D$, which is exclusively caused by generalization. If $\tilde a$ is within a close neighborhood of $a$, then $C_2\|\tilde a-a\|$ is small. Moreover, as $C_1 \in [0,1]$, Eq. (4) approximates an update towards the true objective $\mathcal T_u Q_\theta(s,\tilde a)$, as if $Q_\theta(s,\tilde a)$ is updated by a true gradient step at $(s,\tilde a) \notin \mathcal D$.
>
> We will add these explanations to the latter revision.
>
> **Q3: The meaning of Assumption 1 is unclear at first glance. Rephrase the term "well-defined".**
>
> Thanks for the kind suggestion. Indeed, the term "well-defined" may not be very suitable in this context. To avoid confusion, we will delete the sentence "In other words, $\mathcal{T}_{\mathrm{DMG}}$ is well defined in the mild generalization area $\tilde \beta(a|s)>0$." in Assumption 1. In addition, we will include more explanations before line 158 as follows.
>
> Because the mild generalization area $\tilde \beta(a|s)>0$ may contain some points outside the offline dataset, $\mathcal{T}_{\mathrm{DMG}}$ might query Q values of such points. In this assumption, we assume that the generalization in the mild generalization area is correct and meaningful. The rationale for such an assumption…
>
> **Q4: Minor comments**
>
> Special thanks for your careful review and we will update them in the latter revision.

---

> > ### Comment · Reviewer_RKt1 · 2024-08-09
> >
> > Thank you for your response. Theorem 1 now makes much more sense. Please add that explanation to the main paper.

---

> > > ### Author Response · Authors · 2024-08-09
> > >
> > > Thank you for your valuable suggestions and we will make sure to add the explanations to the main paper. We sincerely appreciate the time and effort you devoted to reviewing our work.

---

### Official Review · Reviewer_kJjy · 2024-07-13

**Soundness:** 3
**Presentation:** 3
**Contribution:** 3
**Rating:** 6
**Confidence:** 5

**Summary:**

This paper studies the problem of extrapolation error and value overestimation in Reinforcement Learning (RL). The authors exploit generalization in offline RL by using mild action generalization and mild generalization propagation.

**Strengths:**

1. The problem studied is on interest : offline RL typically suffers from study of generalization beyond datapoints seen in the dataset.
2. Strong theoretical results complimented with rigorous experiments. I'm impressed with the number of baselines the authors have tried out.

**Weaknesses:**

Assumption 11 looks very restrictive. The scenarios to which the approach can be applied gets significantly reduced if the transition needs to be lipschitz in the action space (for ever state, state next pair). For example for robotics tasks, where transitions are determinisitc, the transition matrix will take values 0-1, mnaking the Lipschitz constant very large in this scenario.

**Questions:**

The idea of generalization to stata-action pairs in the neighborhood of those seen in the dataset has been studied in https://arxiv.org/pdf/2402.12570 and I urge the authors to cite and state the differences to the approach in this paper.

**Limitations:**

This paper has no significant limitations.

---

> ### Author Rebuttal · Authors · 2024-08-07
>
> We appreciate the time and effort you are dedicated to providing feedback on our paper and are grateful for the meaningful comments.
>
> **Q1: Assumption 11 looks very restrictive.**
>
> We apologize for the lack of a detailed explanation of this assumption in the paper. To our knowledge, Assumption 11 (i.e., Assumption 3) is common in the theoretical studies of RL [1,2,3,4,5]. Several previous works assume the transition to be Lipschitz continuous with respect to (w.r.t) both state and action [1,2]. In our paper, we need the Lipschitz continuity to hold only w.r.t. action. However, we also acknowledge, as you mentioned, that the assumption reduces the theoretical applicability. Despite this, DMG performs quite well empirically on those robotics tasks with deterministic dynamics. In real-world scenarios, even for deterministic dynamics, there is some noise, resulting in a narrow distribution for state transitions. In this case, the assumption still applies, although the magnitude of the Lipschitz constant may vary depending on the level of noise.
>
> **Q2: The idea of generalization to stata-action pairs in the neighborhood of those seen in the dataset has been studied in [6] and I urge the authors to cite and state the differences to the approach in this paper.**
>
> Thanks for the reference. The research work [6] is crucial in the analysis of multi-task offline RL, and we will cite it and include the following discussions.
>
> Recently, theoretical advancement [6] has explored multi-task offline RL from the perspective of representation learning and introduced a notion of neighborhood occupancy density. The neighborhood occupancy density at some $(s, a)$ in the dataset for a source task is defined as the fraction of points in the dataset within a certain distance from $(s, a)$ in the representation space. The authors use this concept to bound the representational transfer error in the downstream target task. In contrast, DMG is a wildly compatible idea in offline RL and provides insights into many offline RL methods. DMG balances the need for generalization with the risk of over-generalization in offline RL. Specifically, DMG achieves doubly mild generalization, comprising mild action generalization and mild generalization propagation. Generalization to stata-action pairs in the neighborhood of the dataset is a specific realization of mild action generalization. Additionally, in our work, the neighborhood is defined based on the action space distance.
>
> **Reference**
>
> [1] Dufour, Francois, and Tomas Prieto-Rumeau. "Finite linear programming approximations of constrained discounted Markov decision processes." SIAM Journal on Control and Optimization 51.2 (2013): 1298-1324.
>
> [2] Dufour, Francois, and Tomas Prieto-Rumeau. "Approximation of average cost Markov decision processes using empirical distributions and concentration inequalities." Stochastics An International Journal of Probability and Stochastic Processes 87.2 (2015): 273-307.
>
> [3] Shah, Devavrat, and Qiaomin Xie. "Q-learning with nearest neighbors." Advances in Neural Information Processing Systems 31 (2018).
>
> [4] Xiong, Huaqing, et al. "Deterministic policy gradient: Convergence analysis." Uncertainty in Artificial Intelligence. PMLR, 2022.
>
> [5] Ran, Yuhang, et al. "Policy regularization with dataset constraint for offline reinforcement learning." International Conference on Machine Learning. PMLR, 2023.
>
> [6] Bose, Avinandan, Simon Shaolei Du, and Maryam Fazel. "Offline multi-task transfer rl with representational penalization." arXiv preprint arXiv:2402.12570 (2024).

---

> > ### Comment · Reviewer_kJjy · 2024-08-07
> >
> > I thank the authors for their detailed response. My concerns have been resolved and I've increased my score.

---

> > > ### Author Response · Authors · 2024-08-08
> > >
> > > Thank you for your positive feedback! We truly appreciate your time and effort in reviewing our work.

---

### Author Rebuttal · Authors · 2024-08-07

### **Global Response**

We thank all the reviewers for taking the time to read our manuscript carefully and for providing constructive and insightful feedback. We are encouraged by the positive comments of the reviewers, such as:

- Meaningful research problem, innovative approach, and nice insights (Reviewers kJjy/Kscy/iL9e);
- Strong theoretical results with rigorous proofs (Reviewers kJjy/RKt1/Kscy);
- Superior performance on multiple benchmarks with many baselines and several ablation studies (Reviewers kJjy/RKt1/Kscy/iL9e);
- Seamless transition to online learning and strong fine-tuning performance (Reviewer Kscy).

Meanwhile, we have been working hard to address the reviewers' concerns and questions and have provided detailed responses to the individual reviews below. We hope our response could address the reviewers' concerns. We would be more than happy to resolve any remaining questions in the time we have and are looking forward to engaging in a discussion.

Manuscript updates:

- Update the results with 10 random seeds based on [1].
- Include the discussions regarding [2].
- Add more explanations of how Eq. (4) relates to the generalizability of Q functions.
- Rephrase the term "well-defined" and include more explanations of Assumption 1.
- Correct other typos.

Reference:

[1] Patterson, Andrew, et al. "Empirical design in reinforcement learning." arXiv preprint arXiv:2304.01315 (2023).

[2] Bose, Avinandan, Simon Shaolei Du, and Maryam Fazel. "Offline multi-task transfer rl with representational penalization." arXiv preprint arXiv:2402.12570 (2024).

---

### Decision · Program_Chairs · 2024-09-25

**Decision:**

Accept (poster)

**Comment:**

This paper studies an important problem of generalization in offline reinforcement learning. Recent in-sample algorithms reduce the extrapolation error and value overestimation problem but the resulting policy could be too restrictive. The proposed method, Doubly Mild Generalization (DMG), comprises of mild action generalization and mild generalization propagation. It balances the need for generalization with the risk of over-generalization in offline RL.

All reviewers appreciate its rigorous theoretical analysis and thorough empirical validation on multiple offline RL benchmarks. Some reviewers also consider it a novel approach in offline RL, and did a nice job in explaining the intuition behind the algorithm.

There were some shared concerns on the continuous assumption in the theory, and questions on the sensitivity of hyperparameters, and lack of discusson on related works. The rebuttal has addressed all the concerns. The reviewers reached a unanimous agreement on accepting this submission with a high confidence.